# Temporally Aware Kernel Change-Point Detection with Graph Signal Kernels

## Abstract

Kernel change-point detection methods generally treat the observations of a time series before and after a potential change point as two bags of jumbled i.i.d. samples. They use a kernel to compare these two bags, ignoring the geometric glue that structures the samples in them: the *time* in *time series*. This geometric structure–whether expressed via a graph or via a correlation matrix of spatial or temporal dependencies–is essential for identifying changes that operate not through the mean but spectrally, through changes in the autocorrelative properties of the time series. To make comparisons between time series segments that take these spectral properties into account, we introduce *graph signal kernels*. Whereas graph kernels express relationships on a discrete set (sample times), graph function kernels compare chunks of time series (sample values). Defining graph signal kernels on the RKHS of the graph kernel on our time index set bakes the geometry of the index set into the comparisons between time series chunks. These kernels induce metrics on time series segments that are closely related to the Mahalanobis distance (when covariance matrices encode the structure on the time index set) and Dirichlet metric (when graphs do). They can be directly integrated into classical kernel change-point detection algorithms such as Pruned Exact Linear Time (PELT). Thanks to the fact that the discrete cosine transform (DCT-II) diagonalizes the Laplace matrix of a chain graph, we can efficiently compute kernels between smooth, axis-aligned functions on gridded data using the fast Fourier transform algorithm (FFT). Using time-data tensor product kernels, we can apply this method even when our sample values exist in a large or infinite-dimensional space with or without known geometric structure in the data space.

---

[1]Anonymous Institution, Anonymous City, Anonymous Region, Anonymous Country. Correspondence to: Anonymous Author <anon.email@domain.com>.

To work with irregular grids, we can resample using the time-data tensor product kernel.

## 1. Introduction

In traditional kernel change-point detection (see Fig. 2), the kernel is used to embed individual samples into a reproducing kernel Hilbert space (RKHS; see Appendix A). One transforms a sequence of $T$ vectors of feature *values* $(y_i)_{i=1}^n$, each in $\mathcal{Y} = \mathbb{R}^d$, taken at times $(t_i)_{i=1}^n$ into a sequence $(\phi_{\mathcal{D}}(y_i))_{i=1}^n$ of kernel embeddings of the *values* $y_i$. Here $\phi_{\mathcal{D}}(y_i) = k_{\mathcal{D}}(\cdot, y_i)$ is the Riesz representation of evaluation at the value $y_i$ in the RKHS $\mathcal{D}$ associated with the positive-definite kernel $k_{\mathcal{D}} : \mathcal{Y} \times \mathcal{Y} \to \mathbb{R}$ defined on our index set $\mathcal{Y} = \mathbb{R}^d$ of feature value vectors. With this approach, relationships between times are ignored: the temporal ordering of the sequence is ignored when each cost of a change point is being computed, rendering the method blind to changes in the signal's spectral content.

Whether we use a local heuristic method that tests for a change within a sliding window or a global method that employs dynamic programming to find the lowest-cost set of change points, kernel change point detection always asks the same question (see Fig. 2): Is it worth it to split one bag of kernel embeddings of data values $B = \{\phi_{\mathcal{D}}(y_i)\}_{i=1}^n$ into two, $B_L = \{\phi_{\mathcal{D}}(y_i)\}_{i=1}^t$ and $B_R = \{\phi_{\mathcal{D}}(y_i)\}_{i=t+1}^n$?

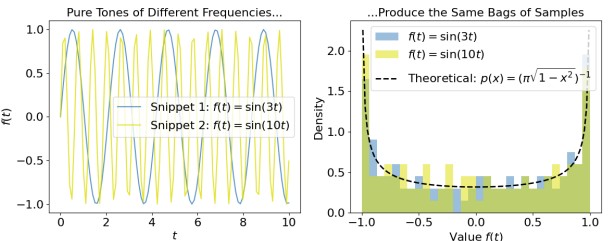

*Figure 1.* The distribution of the values of a pure tone (uniformly sampled) does not depend on its frequency. Left: 100 samples of two tones. Right: the distribution of samples in the associated bag.

Generally, this question is answered using a two-sample statistical test based on the empirical Maximum Mean Discrepancy (MMD) (Arlot et al., 2019; Celisse et al., 2018; Gretton et al., 2012) or an equivalent cost function (see Section E), although scores based on other kernel statistical tests

---

such as the kernel Fisher discriminant ratio are sometimes used (Harchaoui et al., 2008).

If the kernel embedding is injective, this test can, with enough data, detect any difference between the bags. But, as Fig. 1 illustrates, two very different signals can, when dismantled, produce identical bags of unorganized samples.

In this article, we seek to compare not two bags of samples (or their kernel embeddings in $\mathcal{D}$) of a deconstructed time series, but rather two bags of length-$T$ time series *chunks*. These chunks consist not of i.i.d. observations but exhibit correlation due to proximity of the sample locations in a reference index set $\mathcal{T} = \{t_1, \ldots, t_T\}$. The correlative structure on $\mathcal{T}$ is not necessarily determined by proximity in $\mathcal{T}$; it can vary within $\mathcal{T}$ due to some natural phenomenon. For instance, we chunk an EKG signal by heartbeat and resample at times $\mathcal{T}$ with respect to a reference heartbeat, the covariance between two samples $\text{cov}(y_i, y_j)$ depends not just on the distance between their sample locations $t_i$ and $t_j$ in $\mathcal{T}$ but their position within the heartbeat cycle.

**Remark 1.** *Even if we overlap chunks, estimators such as PELT will be consistent under standard assumptions (see Proposition 1). However, with disjoint chunks, a well-chosen chunk size $T$ decorrelates correlated samples into less correlated chunks.*

To make a comparison between time-series chunks (see Fig. 3), we need to use a kernel that is informed by these within-chunk temporal correlations. Whether this correlative structure is specified using a graph or covariance matrix, we call these kernels *graph signal kernels*. This shorthand is purely for expository convenience; many perfectly valid graph signal kernels are specified by covariance matrices $\mathbf{K}$ for which there is no associated graph (i.e., $1_T$ is not an eigenvector of covariance kernel matrix $\mathcal{T}$). Table 1 highlights the difference between graph kernels $k_{\mathcal{T}} : \mathcal{T} \times \mathcal{T} \to \mathbb{R}$–determined by any symmetric, positive semidefinite matrix $\mathbf{K}$ such that $k(t_i, t_j) = \mathbf{K}_{i,j}$–and graph function kernels $k_{\mathcal{X}} : \mathbb{R}^{\mathcal{T}} \times \mathbb{R}^{\mathcal{T}} \to \mathbb{R}$. While a traditional graph signal kernel expresses structural relationship within one single chunk (so that, for instance, smoothing or interpolating graph splines may be constructed), a graph signal kernel measures the similarity between two different chunks of time series or realizations of a single chunk. These notions generalize at once to vector-valued RKHSs.

*Table 1.* Graph kernels versus graph signal kernels. The former express relationships within a chunk; the latter, between chunks.

| KERNEL TYPE | INDEX SET $\mathcal{X}$ | RKHS $\mathcal{H}$ |
| --- | --- | --- |
| GRAPH | VERTICES $\mathcal{T}$ | SIGNALS $\mathbb{R}^{\mathcal{T}}$ |
| GRAPH SIGNAL | SIGNALS $\mathbb{R}^{\mathcal{T}}$ | FUNCTIONS $\mathbb{R}^{\mathcal{T}} \to \mathbb{R}$ |

The remainder of this article is organized as follows. In Sec-

tion 2, we recall the common failure modes of traditional kernel change-point detection methods and remark that the most widely used kernels–i.e., orthogonally-invariant kernels–are insensitive to the rearrangement of samples. Thus, much like traditional kernel change-point detection methods, these kernels, in effect, only consider jumbled bags of samples. Therefore, we must use a different class of kernels to define our graph signal kernels. For this, we turn to an idea from Grace Wahba (Wahba, 1981, and Section C): we synthesize spectrally weighted kernels "on the Fourier side." In doing this, we bake the geometry of $\mathcal{T}$ (expressed using our graph kernel $k_{\mathcal{T}}$) into the graph signal kernel $k_{\mathcal{X}}$ on $\mathcal{X}$ (the RKHS of $k_{\mathcal{T}}$) via the inner product of $\mathcal{X}$. Section 3 provides some examples of simple graph signal kernels on real-valued time series. These kernels use the covariance or graph structure of the discrete domain $\mathcal{T}$ to make geometry- or covariance-respecting comparisons between time series chunks, enabling us to detect spectral differences. In this section, we connect these graph function kernels to the Dirichlet energy and Mahalanobis distance. In Section 4, we use time-data tensor product kernels to adapt these graph function kernels for real-valued time series to times series where high- or infinite-dimensional data are observed at each instant in $\mathcal{T}$. Whether or not this data observation space has known geometric structure, we can embed these observations in an RKHS $\mathcal{D}$. In this case, our graph signal kernels compare embeddings of chunks in a vector-valued RKHS. When such observations are smooth, axis-aligned, and gridded, our time series segment kernels admit implementations using the FFT algorithm. Finally, in Section **??**, we discuss how to implement change-point detection algorithms using time series segment kernels and provide some experimental results.

## 2. Back to Bags: Orthogonally Invariant Kernels Are Insensitive to Jumble

Traditional kernel change-point detection methods fail in many common situations (Van den Burg & Williams, 2020):

- they are insensitive to changes in frequency (see Fig 1);

- they flag many change points when the signal drifts and fail to register changes in the slope of drift;

- they fail to detect changes in the autoregressive structure of a random process; and

- they cannot distinguish between a stationary process and a non-stationary process with the same global energy and marginal distribution.[1]

---

[1] For example, they struggle to detect changes in GARCH processes with constant marginal variance but time-varying conditional variance (Bauwens et al., 2006)).

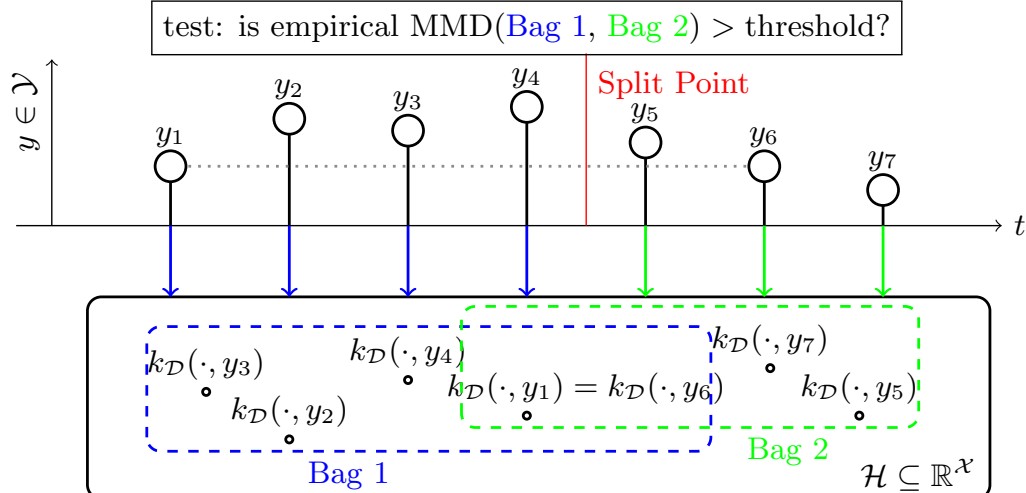

*Figure 2.* Traditional kernel change-point detection uses the empirical MMD between two "bags of samples" (or a related cost) to assess whether to mark a split point. For characteristic kernels, any difference in distribution in the samples of the two bags. However, bags of samples are unordered. When spectral properties change, differences in the distributions of the samples may be incidental. The most prominent differences between the distributions of samples require the order of the observations to be made apparent.

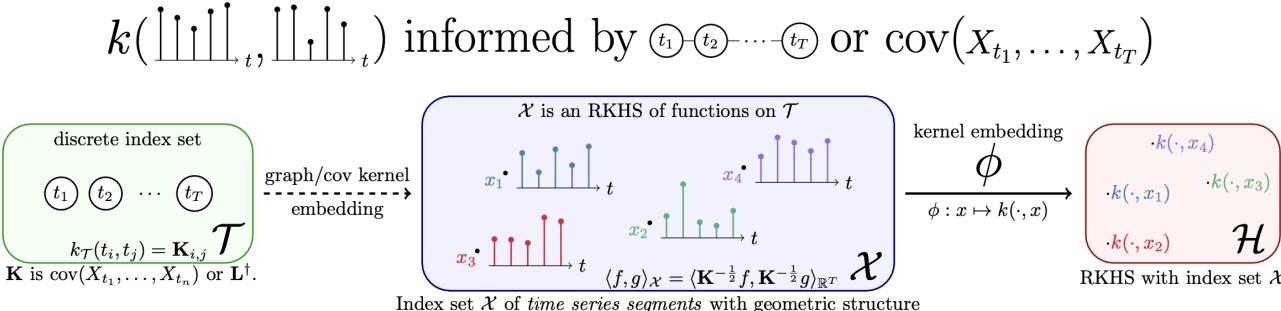

*Figure 3.* Given a graph kernel $k_{\mathcal{T}}$ (specified by a covariance matrix or graph on our index set $\mathcal{T}$), we define a *graph signal kernel* $k_{\mathcal{X}}$ on the RKHS of $k_{\mathcal{T}}$. This facilitates the comparison of *time series chunks* and construction of temporally aware change-point detection methods. That the inner product on $\mathcal{X}$ is determined by the kernel on $\mathcal{T}$ means that common kernel constructions will factor the geometry of $\mathcal{T}$ into our comparisons between chunks of time series. When $\mathbf{K}$ is a covariance matrix or graph Laplace matrix, the linear kernel on $\mathcal{X}$ induces a Mahalanobis or Dirichlet metric on $\mathcal{X}$. When a vector or function $y_i \in \mathcal{Y}$ is observed at each $t_i$ (rather than a scalar $y_i \in \mathbb{R}$), we let $\mathcal{X}$ be the vector RKHS of time series segments of kernel embeddings $\phi_{\mathcal{D}}(y_i) = k_{\mathcal{D}}(\cdot, y_i)$ of these observation vectors.

Such changes involve *spectral properties* and are not preserved when traditional kernel change-point detection methods toss samples into a bag and compute a cost based on the sample kernel embeddings. Traditional kernel change-point detection methods can still detect these changes in spectral properties via concomitant (often more subtle) changes in the distributions of the samples in each bag, or via a dedicated preprocessing chain that encodes these spectral properties into the samples themselves. For example, a difference filter can eliminate false detections in signals with drift (when the drift is constant, the bags of samples are identical) and highlight changes in drift. Note that addressing these failure modes with preprocessing is particularly challenging with time series of vector-valued observations, especially when the autocorrelative properties of only certain features are affected by the changes in the signal.

To perform kernel change-point detection using bags of time series chunks, we require a kernel that is attentive to the spectral properties of the signal chunks being compared. In particular, they must be affected by the action of the orthogonal group $\mathrm{O}(T)$, which notably includes the subgroup of permutations. This excludes most commonly used kernels, which are radial $k(x, y) = \phi(\|x - y\|_{\mathbb{R}^T})$ (e.g., Gaussian, Matérn, Laplacian) or dot-product kernels $k(x, y) = \phi(\langle x, y \rangle_{\mathbb{R}^T})$ (e.g., polynomial, cosine kernel, the neural tangent kernel).

Suppose $k(x, y) = k(\mathbf{Q}x, \mathbf{Q}y)$ for all orthogonal matrices $\mathbf{Q} \in \mathrm{O}(T)$. Since any function invariant under the action of $\mathrm{O}(T)$ is a function of the inner products of its argu-

ments (Weyl, 1946, Theorem 2.9.A, p. 53), it is standard to write $k(x, y) = \kappa(||x||_{\mathbb{R}^T}, ||y||_{\mathbb{R}^T}, \langle x, y \rangle_{\mathbb{R}^T})$. Measures of similarity of bags of samples that pass through such a kernel–such as the MMD–will necessarily be invariant to $\mathbf{Q}$. Since $\mathbf{Q}$ can be a permutation, the change test cannot distinguish between a case where $x$ and $y$ differ (at fixed energy) in scattered indices or in clumped bursts. Since $\mathbf{Q}$ can be a spectral transform such as a unitary DFT matrix[2] $\mathbf{D}_T$, DCT-II matrix, or DWT matrix (using an orthogonal wavelet, such as a Haar or Daubechies db$n$ wavelet, with periodic boundary conditions), discrepancies in low-frequency bins are equivalent to differences in high-frequency bins.

While orthogonally invariant kernels may be suitable for comparing independent realizations of uncorrelated or decorrelated features, such as i.i.d. draws from spherical or standard normal neural latent embeddings, they are inappropriate for correlated draws. When the vector-valued observations $d > 1$ are correlated, the problem compounds: the kernel may also be invariant to permutations of both the feature and time indices. We therefore seek orthogonally sensitive kernels.

The kernels employed in kernel change-point detection libraries are generally orthogonally invariant: `ruptures` (Truong et al., 2018; 2020) and `tadkit` (Confiance.ai, 2024) use linear, Gaussian, and cosine kernels; `kerSeg` (Song & Chen, 2024) and `kcpRS` (Cabrieto et al., 2018), the Gaussian kernel; and `ecp` (James & Matteson, 2014; Matteson & James, 2014) using a distance covariance kernel.[3] One kernel for "chunks" that is not orthogonally invariant is the alignment kernel (Cuturi et al., 2007), which uses a dynamic program to sum over all chunk alignment paths $\exp(-d(x, y))$ where $d$ a conditionally negative definite metric (generally the Euclidean distance or its square).

**Remark 2.** *The orthogonal invariance of traditional kernels can be expressed geometrically. Kernels that cannot distinguish between a signal and its time-reversed, permuted, or frequency-shifted counterparts, provided the global energy remains constant, induce the discrete metric on $\mathcal{T}$. By contrast, orthogonally sensitive kernels on time series chunks induce geometry that connects the elements of $\mathcal{T}$.*

To generate orthogonally sensitive kernels on $\mathcal{X} \subseteq \mathbb{R}^T$, we first choose a kernel $k_{\mathcal{T}} : \mathcal{T} \times \mathcal{T} \to \mathbb{R}$, which can be identified with any $T \times T$ symmetric, positive-semidefinite matrix $\mathbf{K}$. Following the approach of (Wahba, 1981, Appendix C) used by Wahba to synthesis interpolating splines, we

[2]Or, to keep it real, $\mathfrak{Re}\{\mathbf{D}_T\} - \mathfrak{Im}\{\mathbf{D}_T\}$, which also diagonalizes the length-$T$ cycle graph.

[3]The one exception we found: the repository `chapydette` (Jones & Harchaoui, 2020) uses Gaussian kernels defined on metrics from information geometry, such as the Hellinger distance and $\chi^2$ kernel; the latter is not orthogonally invariant. Nevertheless, each test, performed on bags of samples, is permutation invariant.

synthesize $\mathbf{K}$ using a basis $\mathbf{U} \in O(d)$ derived from a covariance matrix (i.e., the principal components) or a graph (the Laplace matrix eigenvectors) and a diagonal matrix of weights $\mathbf{\Lambda} = \mathrm{diag}(\lambda_1, \ldots, \lambda_T) \succeq 0$.

We shall then depart from Wahba's work to use these kernels $k_{\mathcal{T}}$, which specify similarity on the index set, to define kernels $k_{\mathcal{X}}$, which compare the values of a time series on chunks. We summarize the two families of kernels $k_{\mathcal{T}}$ used in this work in the following box.

---

**Generating K: Dirichlet or Mahalanobis**

Given a graph Laplace matrix $\mathbf{L}$ or covariance matrix $\mathbf{C}$, we set $\mathbf{K} = \mathbf{L}^\dagger$ or $\mathbf{K} = \mathbf{C}$, where $\dagger$ is the Moore-Penrose pseudoinverse. Using the spectral decomposition $\mathbf{K} = \mathbf{U}\mathbf{\Lambda}\mathbf{U}^T$, we define, via spectral transformation, a family of kernels on $\mathcal{T}$, $k_{\mathcal{T}} : \mathcal{T} \times \mathcal{T} \to \mathbb{R}$:

$$k_{\mathcal{T}}(t_i, t_j) \stackrel{\text{def}}{=} \psi(\mathbf{K})_{j,l} = \mathbf{U}\psi(\mathbf{\Lambda})\mathbf{U}^T = \sum_{i=1}^{T} \psi(\lambda_i)u_i(j)u_i(l);$$

the scalar function $\psi$ acts elementwise on eigenvalues. Some natural choices of $\psi$ include the following:

- Green's: $\psi_{\mathrm{g}}(\lambda) \stackrel{\text{def}}{=} \lambda$;

- Sobolev: $\psi_{\mathrm{s}}(\lambda) \stackrel{\text{def}}{=} 1 + \lambda$;

- Thin-plate spline: $\psi_{\mathrm{tps}}(\lambda) \stackrel{\text{def}}{=} \lambda^m$ with $m > 0$.

- Heat: $\psi_{\mathrm{h}}(\lambda) \stackrel{\text{def}}{=} \exp(\tau\lambda)$ for some $\tau > 0$.

The RKHS $\mathcal{X}$ of functions $\mathbb{R}^T \times \mathbb{R}^T \to \mathbb{R}$ associated with the kernel $k_{\mathcal{T}}$ is $\mathcal{X} = \mathrm{span}\{u_i \mid \psi(\lambda_i) > 0\}_{i=1}^{T}$ with inner product $\langle x, y \rangle_{\mathcal{X}} = x^T\psi(\mathbf{K})^\dagger y$.

---

We confirm that the kernel embedding of $t_i \in \mathcal{T}$ is $\phi_{\mathcal{T}}(t_i) \stackrel{\text{def}}{=} k_{\mathcal{T}}(\cdot, t_i) = \psi(\mathbf{K})[:, i]$:

$$\langle \phi_{\mathcal{T}}(t_i), \phi_{\mathcal{T}}(t_j) \rangle_{\mathcal{X}} = e_i^T\psi(\mathbf{K})^T\psi(\mathbf{K})^\dagger\psi(\mathbf{K})e_j = \psi(\mathbf{K})_{i,j}.$$

For all $f \in \mathcal{X}$, $\langle f, \phi_{\mathcal{T}}(t_i) \rangle_{\mathcal{X}} = f^T\psi(\mathbf{K})^\dagger\psi(\mathbf{K})e_i = f(t_i)$.

## 3. Graph Signal Kernels Derived from Graph Laplace Matrices and Covariance Matrices

The kernel $k_{\mathcal{T}}$ compares *sample locations* in $\mathcal{T}$; it encodes its "geometry." This kernel is completely determined by the Gram matrix $\psi(\mathbf{K})$ (which we shall simply refer to as $\mathbf{K}$ when we do not wish to emphasize that spectral transformations may be used in its construction) defined by $\mathbf{K}_{i,j} = k_{\mathcal{T}}(t_i, t_j)$. Associated with this kernel is an RKHS $\mathcal{X}$ of time series segments. We shall use this space to define a kernel for graph signals.

We begin with the case where $\mathcal{D} = \mathbb{R}$. Thus, our graph

signals are functions $\mathbb{R}^{\mathcal{T}}$ on this discrete set, each of which can be identified with a vector of evaluations in $\mathbb{R}^T$. We consider the simplest kernel on these time series chunks: the linear kernel.

**Example 1** (Linear kernel). *Let* $\mathbf{K} = \mathbf{U}\mathbf{\Lambda}\mathbf{U}^T$ *be the Gram matrix of* $k_{\mathcal{T}}$ *with* $u_i = \mathbf{U}[:, i]$ *and* $\mathbf{\Lambda} = diag(\lambda_1, \ldots, \lambda_T) \succeq 0$. *The space* $\mathcal{X} = \mathrm{span}\{u_i \,|\, \lambda_i > 0\}_{i=1}^T$ *is an RKHS of length-T, real-valued time series segments. Using the RKHS* $\mathcal{X}$ *as an index set, the linear kernel* $k_{\mathcal{X}} : \mathcal{X} \times \mathcal{X} \to \mathbb{R}$ *can be written as follows:*

$$k_{\mathcal{X}}(x, y) = \langle x, y \rangle_{\mathcal{X}} = \sum_{\substack{i=1 \\ \lambda_i > 0}}^T \frac{\langle x, u_i \rangle_{\mathbb{R}^T} \langle y, u_i \rangle_{\mathbb{R}^T}}{\lambda_i}$$

$$= \langle \mathbf{W}x, \mathbf{W}y \rangle_{\mathbb{R}^T} = x^T \mathbf{K}^{\dagger} y,$$

*where* $\mathbf{W} = \mathbf{\Lambda}^{\dagger/2}\mathbf{U}^T$. *The RKHS associated with* $k_{\mathcal{X}}$ *is the continuous dual space* $\mathcal{H} = \mathcal{X}^*$ *of* $\mathcal{X}$. *By the Riesz representation theorem, the kernel embedding of any time series chunk* $x \in \mathcal{X}$ *is* $\phi_{\mathcal{X}}(x) = \langle \cdot, x \rangle_{\mathcal{H}}$ *and the reproducing property holds.*

**Remark 3.** *The matrix* $\mathbf{W}$ *is a whitening filter when* $\mathbf{K} = \mathbf{C}$ *and it decorrelates AR(1) processes. The norm is Fourier-weighted: it excludes the contribution of modes with* $\lambda_i = 0$ *to the signal energy and downweights the signal energy modes associated with large values of* $\lambda_i$*–generally "nice" modes (high explanation-of-variance when* $\mathbf{K} = \mathbf{C}$ *or low Dirichlet energy when* $\mathbf{K} = \mathbf{L}^{\dagger}$*). With uniform weighting* $\lambda_1 = \ldots = \lambda_T$, $k_{\mathcal{X}}$ *becomes orthogonally invariant.*

**Remark 4** (The linear kernel determines the Dirichlet and Mahalanobis metrics). *When* $k$ *is a positive-definite kernel and* $\mathcal{H}$ *its RKHS, the pseudometric[4]* $d$ *satisfying*

$$d(f, g)^2 = ||f - g||_{\mathcal{H}}^2 = k(f, f) + k(g, g) - 2k(f, g)$$

*is definite and thus a metric. When* $k$ *is the linear kernel and* $\mathbf{K}$ *a covariance matrix,* $d$ *is the Mahalanobis distance. When* $k$ *is the linear kernel and* $\mathbf{K}$ *the Moore-Penrose pseudoinverse of a graph Laplace matrix, then* $d$ *is the Dirichlet metric on zero-mean graph functions; when the Decomposition Principle (see Remark 7) is used, it becomes the standard discrete* $\mathcal{H}^1$ *Sobolev metric.*

While Fig. 4 suggests that even a linear kernel on $\mathcal{X}$ is sensitive to spectral changes, the linear kernel is highly restrictive. Its associated RKHS is $T$-dimensional and consists of only

---

[4]For a general RKHS $\mathcal{H}$ associated with positive-definite kernel $k$ on index set $\mathcal{X}$, the function $d(x, y) = ||k(\cdot, x) - k(\cdot, y)||_{\mathcal{H}}$ is a *pseudometric*. Strict positive-definiteness of $k$ is sufficient (but not necessary) to enforce definiteness: when $x \neq y$, the Gram matrix of $k$ on $\{x, y\}$ is strictly positive-definite so its determinant $k(x, x)k(y, y) - k(x, y)^2 > 0$. By the inequality of arithmetic and geometric means, this implies $d(x, y) = (k(x, x) + k(y, y) - 2k(x, y))^{\frac{1}{2}} > 0$; thus, $d$ is a metric.

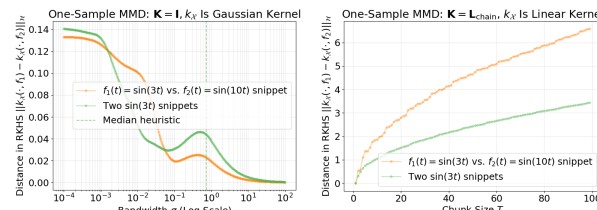

*Figure 4.* Distance in $\mathcal{H}$ of the embeddings of $\sin(3t)$ and $\sin(10t)$ pure tone samples of Fig. 1. Left: with $k_{\mathcal{X}}(t_i, t_j) = \delta_{ij}$, these different frequencies are closer in $\mathcal{H}$ than two snippets of $\sin(3t)$ (with slight phase difference) over many bandwidths, including at the median heuristic. Right: setting $\mathbf{K} = \mathbf{L}^{\dagger}$, where $\mathbf{L}$ is the chain graph Laplacian, the linear kernel can distinguish tones of different frequencies even at small chunk sizes.

the linear functionals $\langle \cdot, x \rangle_{\mathcal{X}}$, where $x$ is some length-$T$ time series segment in $\mathcal{X}$. Other positive-definite kernels on $\mathcal{X}$, such as the Gaussian kernel and Laplacian kernel, have as their RKHS more expressive, infinite-dimensional spaces of functions $f : \mathcal{X} \to \mathbb{R}$. These two kernels are examples of universal kernels: kernels whose associated RKHSs $\mathcal{H}$ are so rich that arbitrary continuous target functions $f : \mathbb{R}^T \to \mathbb{R}$ can be approximated uniformly by an element of $\mathcal{H}$ on any compact subset of $\mathcal{X} \cong \mathbb{R}^T$ (Micchelli et al., 2006).

**Example 2** (Gaussian kernel, Laplacian kernel, . . . ). *The Gaussian kernel, with bandwidth* $\rho$, *can be written*

$$k_G^{\rho}(x, y) = \exp\left(-\frac{||x - y||_{\mathcal{X}}^2}{2\rho^2}\right) = \prod_{\substack{i=1 \\ \lambda_i > 0}}^T \exp\left(\frac{\langle x - y, u_i \rangle_{\mathbb{R}^T}^2}{2\rho^2 \lambda_i}\right)$$

$$= \exp\left(-\frac{(x - y)^T \mathbf{K}^{\dagger}(x - y)}{2\rho^2}\right);$$

*it is proportional to the Gaussian likelihood ratio between two segments.*

*The Laplacian kernel can be defined as follows:*

$$k_L^{\alpha}(x, y) = \exp\left(-\alpha ||x - y||_{\mathcal{X}}\right).$$

*We have simply replaced the norm* $||x - y||_{\mathbb{R}^T}$ *in the usual definition of these kernels with* $||x - y||_{\mathcal{X}}$. *While swapping Euclidean norms with other norms is treacherous, the kernel remains positive-definite since* $||x - y||_{\mathcal{X}} = ||\mathbf{W}(x - y)||_{\mathbb{R}^T}$ *with* $\mathbf{W} = \mathbf{\Lambda}^{\dagger/2}\mathbf{U}^T$.

## 4. Vector-Valued Graph Signal Kernels

We use a generalized Dirichlet energy and generalized Mahalanobis distance to motivate our graph signal kernels for vector observations.

### 4.1. Generalized Dirichlet Energy

In this section, we use a graph $G = (V, E)$ to impose structure on $\mathcal{T}$. For exposition clairty, we use the labeling isomorphism to identify each $t_i \in \mathcal{T}$ and the corresponding vertex in $V$ with $i \in \{1, \ldots, T\}$ Each edge $\{i, j\} \in E$ has nonnegative weight $w_{ij}$. A scalar-valued graph signal $f(i) = y_i \in \mathbb{R}$, identified with a vector $f \in \mathbb{R}^T$, has Dirichlet energy

$$E(f) = \sum_{\substack{e \in E \\ e = \{i, j\}}} w_{i,j} ||f(i) - f(j)||_{\mathbb{R}}^2 = f^T \mathbf{L} f.$$

If, at each vertex $i$, $f$ assumes not a scalar value but rather a kernel embedding the observation $y_i$–i.e., $f(i) = \phi_{\mathcal{D}}(y_i) = k_{\mathcal{D}}(\cdot, y_i) \in \mathcal{D}$–its generalized Dirichlet energy becomes (see Appendix G):

$$E(f) = \sum_{\substack{e \in E \\ e = \{i, j\}}} w_{i,j} ||f(i) - f(j)||_{\mathcal{D}}^2$$

$$= \sum_{i=1}^{T} \sum_{j=1}^{T} \mathbf{L}_{i,j} \langle f(i), f(j) \rangle_{\mathcal{D}} = \langle \mathbf{L}, \mathbf{G} \rangle_F \quad (1)$$

where $\mathbf{G}$ is the Gram matrix of $\langle \cdot, \cdot \rangle_{\mathcal{D}}$ on the functions $\{f(i)\}_{i=1}^{T}$, satisfying $\mathbf{G}_{i,j} = \langle f(i), f(j) \rangle_{\mathcal{D}}$. Observe that, when $\mathcal{Y} = \mathbb{R}$ and the embeddings $\phi_{\mathcal{D}}(y_i)$ are trivial $\langle \cdot, y_i \rangle_{\mathcal{Y}}$, we recover the usual Dirichlet energy $f^T \mathbf{L} f = \text{trace}(\mathbf{L} f f^T)$. We recall that the inner product $\langle f, g \rangle_{\mathcal{X}} = f^T \mathbf{L} g = \langle \mathbf{\Lambda}^{\dagger/2} f, \mathbf{\Lambda}^{\dagger/2} g \rangle_{\mathbb{R}^T}$ was the inner product on $\mathcal{X}$ for scalar time series chunks. We can obtain a similar result for chunks of RKHS-embedding-valued time series.

In the continuous limit, where the graph becomes a manifold $\mathcal{M}$, we get an energy of the embedding $f(x) = \phi_{\mathcal{D}}(y(x))$

$$E(f) \int_{\mathcal{M}} \text{trace}_{\mathcal{D}}(\nabla f(x)^* \nabla f(x)) \text{dvol}(x),$$

which sums up the energy expended when embedding the data observations $y(x)$ into the RKHS $\mathcal{D}$. Working in infinite-dimensional spaces generally requires a great deal of care, but thanks to our finite observations and the Wahba-Kimeldorf representer theorem, we only need bookkeeping.

Suppose $f = (\phi_{\mathcal{D}}(y_1), \ldots, \phi_{\mathcal{D}}(y_T))^T \in \mathcal{D}^T$ is a vector-valued function on the vertex set. Thus, the Gram matrix $\mathbf{G}$ satisfies $\mathbf{G}_{i,j} = \langle f(i), f(j) \rangle_{\mathcal{D}} = \langle \phi_{\mathcal{D}}(y_i), \phi_{\mathcal{D}}(y_j) \rangle_{\mathcal{D}}$. The rank of $\mathbf{G}$ can be up to $T$: each dimension of $\mathcal{D}$ can enrich the relationship between the $T$ sample locations of our data. We start from (1) and we use the fact that $\mathbf{G}$–being a symmetric, positive semidefinite matrix–can be written $\mathbf{G} = \mathbf{\Phi} \mathbf{\Phi}^T$. This can be accomplished, for instance, using the Cholesky decomposition $\mathbf{G} = \mathbf{\Phi} \mathbf{\Phi}^T$, although this is not unique: for any orthogonal matrix $\mathbf{Q} \in \text{O}(\text{rank}(\mathbf{G}))$, setting $\mathbf{M} = \mathbf{\Phi} \mathbf{Q}$ also yields $\mathbf{G} = \mathbf{M} \mathbf{M}^T$. (We could also

approximate this using random Fourier features (Rahimi & Recht, 2007).)

Let $\mathcal{S} = \text{span}\{\phi_{\mathcal{D}}(y_i)\}_{i=1}^{T}$. Then $\mathcal{S}$ is a subspace of $\mathcal{D}$ of dimension $d' \leq T$. Choose an orthonormal basis $(\xi_i)_{i=1}^{d'}$ (with respect to the inner product $\langle \cdot, \cdot \rangle_{\mathcal{D}}$) of $\mathcal{S}$. We use the Fourier analysis operator

$$A : \mathcal{S} \to \mathbb{R}^{d'}$$
$$f \mapsto \alpha = (\langle f, \xi_1 \rangle_{\mathcal{D}}, \ldots, \langle f, \xi_{d'}, \rangle_{\mathcal{D}})^T$$

to associate embeddings $\phi_{\mathcal{D}}(y_i)$ of data observations in $\mathcal{S} \subseteq \mathcal{D}$ with vectors in $\mathbb{R}^T$. The operator A is an isometry, converting an inner product on $\mathcal{S} \subseteq \mathcal{D}$ into one on $\mathbb{R}^{d'}$:

$$\begin{aligned}
\mathbf{G}_{i,j} &= \langle \phi_{\mathcal{D}}(y_i), \phi_{\mathcal{D}}(y_j) \rangle_{\mathcal{D}} \\
&= \left\langle \sum_{l=1}^{d'} \langle \phi_{\mathcal{D}}(y_i), \xi_l \rangle_{\mathcal{D}} \xi_l, \sum_{l=1}^{d'} \langle \phi_{\mathcal{D}}(y_j), \xi_l \rangle_{\mathcal{D}} \xi_l \right\rangle_{\mathcal{D}} \\
&= \sum_{l=1}^{d'} \langle \phi_{\mathcal{D}}(y_i), \xi_l \rangle_{\mathcal{D}} \langle \phi_{\mathcal{D}}(y_j), \xi_l \rangle_{\mathcal{D}} \\
&= \langle A\phi_{\mathcal{D}}(y_i), A\phi_{\mathcal{D}}(y_j) \rangle_{\mathbb{R}^{d'}}. \quad (2)
\end{aligned}$$

Construct the $T \times d'$ matrix $\mathbf{\Phi}$ by placing in its $i$th row, for $i \in \{1, \ldots, T\}$, the vector $A\phi_{\mathcal{D}}(y_i)$. Each column represents a different "principal feature" of the signal chunk. Then, by Equation (2), $\mathbf{G} = \mathbf{\Phi} \mathbf{\Phi}^T$ and we obtain the following expression for the Dirichlet energy (1):

$$\langle \mathbf{L}, \mathbf{G} \rangle_F = \text{trace}(\mathbf{L} \mathbf{G}) = \text{trace}\left(\mathbf{\Phi}^T \mathbf{L} \mathbf{\Phi}\right) = ||\mathbf{\Lambda}^{1/2} \mathbf{U}^T \mathbf{\Phi}||_F^2$$

By the construction of $\mathbf{\Phi}$ with the orthonormal basis $(\xi_i)_{i=1}^{d'}$, this is a decorrelating filter: the energy $\text{trace}(\mathbf{\Phi}^T \mathbf{L} \mathbf{\Phi})$ generalizes the Dirichlet energy $f^T \mathbf{L} f$ *while accounting for feature correlation*. The $l$th column of $\mathbf{\Phi}$ is the length-$T$ vector $\alpha_l$ of weights on orthogonal feature $\xi_l$ of the Fourier expansion of the kernel embeddings $\phi_{\mathcal{D}}(y_i)$ of the vector observations $y_i$, one for each sample location in $\mathcal{T}$:

$$\phi_{\mathcal{D}}(y_i) = \sum_{l=1}^{d'} \alpha_l^{(i)} \xi_l = \sum_{l=1}^{d'} \langle \phi_{\mathcal{D}}(y_i), \xi_l \rangle_{\mathcal{D}} \xi_l.$$

The Dirichlet energy may be expressed in terms of the Fourier expansion weight vectors $\alpha_l$:

$$\langle \mathbf{L}, \mathbf{G} \rangle_F = ||\mathbf{\Lambda}^{1/2} \mathbf{U}^T \mathbf{\Phi}||_F^2 = \sum_{i=1}^{T} \sum_{l=1}^{d'} \lambda_i \langle u_i, \alpha_l \rangle_{\mathbb{R}^T}^2,$$

where $\alpha_l = (\langle \phi_{\mathcal{D}}(y_1), \xi_l \rangle_{\mathcal{D}}, \ldots, \langle \phi_{\mathcal{D}}(y_T), \xi_l \rangle_{\mathcal{D}})^T$.

### 4.2. Generalized Dirichlet Kernels

When we observe vectors $y_i \in \mathcal{Y} = \mathbb{R}^d$ at each time $t_i$ and embed them into $\mathcal{D}$ using $\phi_{\mathcal{D}}(y_i) = k_{\mathcal{D}}(\cdot, y_i)$, the index

set for our graph signal kernel comparing the embeddings $f = (\phi_{\mathcal{D}}(x_i))_{i=1}^T$ and $g = (\phi_{\mathcal{D}}(y_i))_{i=1}^T$ of two chunks $\mathbf{X} \in \mathbb{R}^{T \times d}$ and $\mathbf{Y} \in \mathbb{R}^{T \times d}$ becomes $\mathcal{X} = \overline{\mathbb{R}^T} \otimes \mathcal{D}$, where the zero-mean vectors $\overline{\mathbb{R}^T} \overset{\text{def}}{=} \text{span}(1_T)^\perp$ form the RKHS of $k_{\mathcal{D}}$ when $\mathbf{K} = \mathbf{L}^\dagger$. The Dirichlet kernel becomes

$$k_{\mathcal{X}} : (\overline{\mathbb{R}^T} \otimes \mathcal{D}) \times (\overline{\mathbb{R}^T} \otimes \mathcal{D}) \to \mathbb{R}$$

$$((\phi_{\mathcal{D}}(x_i))_{i=1}^T, (\phi_{\mathcal{D}}(y_i))_{i=1}^T) \mapsto \text{trace}\,(\mathbf{LG})\,, \text{ where}$$

$\mathbf{G}^c$ is the cross-Gram matrix: $\mathbf{G}_{i,j}^c = \langle \phi_{\mathcal{D}}(x_i), \phi_{\mathcal{D}}(y_j) \rangle_{\mathcal{D}}$. It is a positive-definite kernel since it is the inner product of the tensor product of two Hilbert spaces is a new Hilbert space with a valid inner product (Micchelli & Pontil, 2004).

Equivalently, we may write

$$k_{\mathcal{X}} : (\mathbb{R}^T \otimes \mathcal{D}) \times (\mathbb{R}^T \otimes \mathcal{D}) \to \mathbb{R}$$

$$((\phi_{\mathcal{D}}(x_i))_{i=1}^T, (\phi_{\mathcal{D}}(y_i))_{i=1}^T) \mapsto \text{trace}\,(\mathbf{LHG}^c\mathbf{H})\,,$$

where $\mathbf{H} = \mathbf{I} - 1_T 1_T^T$ is the centering operator. Using the spectral transformation functions defined in Section 2, we obtain the family of kernels $k(f, g) = \text{trace}(\psi(\mathbf{L})\mathbf{HG}^c\mathbf{H})$.

The induced metric is the Dirichlet metric:

$$d_{\mathcal{X}}(f, g) = \|f - g\|_{\mathcal{X}}^2 = (f - g)^T \mathbf{L}(f - g).$$

### 4.3. Generalized Mahalanobis Kernels

The squared Mahalanobis distance

$$d_M(v, \mathcal{N}(\mu, \mathbf{C}))^2 = (v - \mu)^T \mathbf{C}^\dagger (v - \mu)$$

is typically used to compare a vector $v \in \mathbb{R}^d$ to a multivariate Gaussian distribution $\mathcal{N}(\mu, \mathbf{C})$ (or an empirical estimate thereof), where $\mu \in \mathbb{R}^d$ is the mean vector and $\mathbf{C} \in \mathbb{R}^{d \times d}$ the covariance matrix. It is closely related to the square root of the negative log likelihood $-\log p_{v \sim \mathcal{N}(\mu, \mathbf{C})}(v)$. We can also define a squared Mahalanobis distance between two vectors:

$$d_M(x, y; \mathbf{C}) = (x - y)^T \mathbf{C}^\dagger (x - y) = \|\mathbf{C}^{\dagger/2}(x - y)\|_{\mathbb{R}^d}^2$$

$$= \|\mathbf{U}\boldsymbol{\Lambda}^{\dagger/2}\mathbf{U}^T(x - y)\|_{\mathbb{R}^d}^2$$

$$= \|\mathbf{Q}\boldsymbol{\Lambda}^{\frac{\dagger}{2}}\mathbf{U}^T(x - y)\|_{\mathbb{R}^d}^2 \,\, \forall \mathbf{Q} \in \mathrm{O}(d),$$

since orthogonal matrices are isometries. We remark that $\mathbf{W} = \boldsymbol{\Lambda}^{-1/2}\mathbf{U}^T$ is the standard PCA whitening matrix and $\{\mathbf{Q}\boldsymbol{\Lambda}^{-1/2}\mathbf{U}^T \mid \mathbf{Q} \in \mathrm{O}(d)\}$ is the space of all whitening matrices (Kessy et al., 2018, Equation (4)).

Along these lines, we can define the Mahalanobis distance between kernel embeddings in an RKHS $\mathcal{D}$ with index set $\mathcal{Y}$. Since the covariance operator $\Sigma_P$ (see A) is compact, its eigenvalues accumulate at 0 and its inverse is unbounded. The simplest way to proceed is to use Tikhonov regularization with some $\lambda > 0$:

$$d_M(f, g)^2 = \langle f - g, (\Sigma_P + \lambda I)^{-1}(f - g) \rangle_{\mathcal{D}}.$$

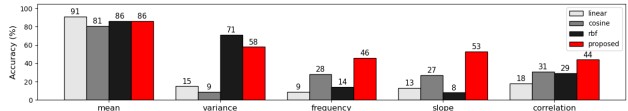

*Figure 5.* Accuracy comparison of 4 models on 500 signals (length 200) with 5 change types and a single changepoint at random $t^* \in [80, 120]$, where a detection $\hat{t}$ is correct if $\hat{t} \in (t^*-5, \, t^*+5)$.

It accentuates differences between $f$ and $g$ that do not align with the variance of $P$. We prefer the centered covariance operator $\overline{\Sigma_P} : \mathcal{D} \to \mathcal{D}$:

$$f \mapsto \int_{\mathcal{Y}} (f - \mu_P) \otimes (f - \mu_P) \mathrm{d}P(x).$$

We move to our vector RKHS $\mathcal{X} \otimes \mathcal{D}$, where $\mathcal{X}$ is the RKHS of $k_{\mathcal{T}}$. Define the $T \times T$ block covariance operator $\Sigma : (\mathcal{X} \otimes \mathcal{D}) \to (\mathcal{X} \otimes \mathcal{D})$ whose $(i, j)$th block contains the cross-covariance operator between the $i$th and $j$th position in two sequences: $\mathbb{E}_P[(\phi_{\mathcal{D}}(x_i)]$. We assume a separable covariance structure, so that the joint precision operator decomposes as $\mathbf{K}^\dagger \otimes \mathrm{I}_{\mathcal{D}}$. Thus, the temporal correlation is fixed by $\mathbf{K}$; $k_{\mathcal{D}}$ handles feature correlations.

Let $f = (\phi_{\mathcal{D}}(x_i))_{i=1}^T$ and $g = (\phi_{\mathcal{D}}(y_i))_{i=1}^T$ be two vectors of embeddings. Let $\xi_{i,j} = \phi_{\mathcal{D}}(x_i) - \phi_{\mathcal{D}}(y_j)$. Then

$$d_M(f, g)^2 = \sum_{i=1}^T \sum_{j=1}^T \mathbf{K}_{i,j}^\dagger \langle \xi_{i,j}, (\xi_{i,j}) \rangle_{\mathcal{D}}.$$

In the general case, we can write

$$d_M(f, g)^2 = \sum_{i=1}^T \sum_{j=1}^T \mathbf{K}_{i,j}^\dagger \langle \xi_{i,j}, \Sigma_{\mathcal{D}}^{-1}(\xi_{i,j}) \rangle_{\mathcal{D}}.$$

This expression decorrelates the signal in both time and along the feature axis: $\mathbf{K}^\dagger$ discounts changes that align with the natural drift of the process, while $\Sigma_{\mathcal{D}}^{-1}$ accounts for the feature-wise correlations.

We now arrive at the Mahalnobis kernel, which is the standard (centered) time-data product kernel:

$$k_{\mathcal{X}}(f, g) = \text{trace}\,(\mathbf{KHG}^c\mathbf{H})\,, \tag{3}$$

where $\mathbf{G}^c$ remains the cross-Gram matrix. We can extend this to $\mathbf{G}_{\mathrm{R}}^c$ whose $(i, j)$th element contains $\langle \phi_{\mathcal{D}}(x_i), \mathrm{R}\phi_{\mathcal{D}}(y_j) \rangle_{\mathcal{D}}$. When $\mathrm{R} = \Sigma_{\mathcal{D}}^{-1}$, the kernel emphasizes "surprising" spectral differences, i.e., those that deviate from the expected covariance structure of the signal.

## 5. Experiments

This article introduces a new framework for generating temporally aware kernels that can be plugged directly into existing kernel change-point detection algorithms, such as binary

segmentation and PELT. See Appendix E for an review of the loss and proof of consistency even in the case where chunks overlap.

These kernels are highly configurable and adaptable to different types of data. The Gram matrix $\mathbf{K}$ of the kernel $k_\mathcal{T}$, in particular, should be adapted to the correlative structure of the data. The chain graph Laplace matrix $\mathbf{L}$ can be profitably used as a default, but many applications involve data with known correlative structure: e.g., satellite brightness temperature data vary faster at high incidence angle, following Fresnel's law; certain seasonal signals such as solar power generation vary faster at the equinox than solstice; and EKG signals follow a P-QRS-T template. In such cases, $\mathbf{K}$ can be modeled directly or empirically with a covariance matrix $\mathbf{C}$. In online setttings, adaptive estimates may be used.

While many researchers propose protocols for evaluating change-point detections (Van den Burg & Williams, 2020), change-point detection practitioners continue to rely on qualitative analyses. The active debate about metrics and signals (Truong et al., 2020) is often divorced from practitioners' varied needs. While the qualitative analyses that dominated unreproducible research articles in the past made researchers' and practitioners' lives difficult, recent focus on benchmarks of a task for which human annotators rarely reach consensus does not simplify matters (Munroe, 2011). Modern reproducible research practices (Colom et al., 2018) allow practitioners to "just try it."

In that spirit, we encourage the reader to play with a working demo, requiring no code, available at Image Processing On-Line. It offers the following options for constructing $k_\mathcal{X}$:

- chunk size $T$;

- base kernel $\mathbf{K} = \mathbf{C}$ (covariance; imported from file) or $\mathbf{K} = \mathbf{L}^\dagger$ (Laplace; chain graph used);

- Spectral transformation parameter $\tau > 0$ and family $\psi = \mathrm{id}$ (Green's) $\psi(\lambda) = \tau + \lambda$ (Sobolev), $\psi(\lambda) = \lambda^\tau$ (TPS), $\psi(\lambda) = \exp(\tau\sigma^2)$ (heat); and

- choice of $k_\mathcal{X}$: linear, Gaussian, or Laplace kernel on the RKHS $\mathcal{X}$ of $k_\mathcal{T}$.

Additionally, the user can choose between kernel binary segmentation and kernel PELT.

Some illustrative examples involving changes in mean, variance, frequency, slope, and autocorrelation are provided in Figures 6-14. See Appendix (**?**).

As Fig. 6 illustrates, chunk size $T$ should be large enough to capture the signal's spectral dynamics but small enough to maintain the resolution of the change points (especially if there is stride used).

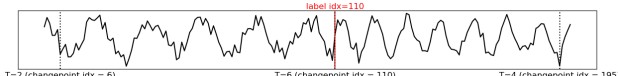

*Figure 6.* $T$ has to be big enough to register the spectral properties that change. With $T > 6$ and stride of 1 (overlapping chunks), there was not much jitter in the detection.

## 6. Future Work

The main challenge to deployment is setting the $T$. Future work will seek to determine reasonable choices of chunk size $T$ from the DCT-II spectral decay of chunks or empirical chunk covariance matrices. While Laplace matrix-based graph signal kernels are well-suited to online change-point detection, integrating an adaptive covariance matrix into online methods will require care: the tool with which we detect the change moves with the underlying change. Extending these kernels more gracefully to the case of irregularly sampled data, to enable the comparison two time series chunks of scattered samples without resampling, is also of interest.

## Acknowledgements

Masked for anonymity.

## Code Availability

We will publish a notebook with the results on GitHub and a full code and demo description submission (which, sadly, is necessarily de-anonymized) to accompany our anonymous demo, currently available on the reproducible research website Image Processing On-Line, after de-anonymization. In the mean time, the pertinent code is placed in Appendix H.

## Impact Statement

In addition to providing researchers a new framework for analyzing kernel change-point detection methods, this work seeks to make time series change-point detection methods more effective and to reduce the need for developing specialized processing chains that render changes in spectral properties apparent to traditional methods. We imagine this work to be most useful to practitioners with limited computational resources and expertise in signal processing who seek simple and accessible methods for kernel change-point detection for "one-off tasks" in data annotation and analysis. Substantial societal impacts are unlikely to emerge from this work: high-stakes applications demand tailored processing chains or neural networks.

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

## Appendices

## A. RKHS, Mean Embeddings, and All That

To every positive-definite kernel $k : \mathcal{X} \times \mathcal{X} \to \mathbb{R}$ is associated a unique reproducing kernel Hilbert space (RKHS) $\mathcal{H}_k$, a Hilbert space of functions from $\mathcal{X}$ to $\mathbb{R}$ such that the linear evaluation functional $\mathrm{E}_x : \mathcal{H}_k \to \mathbb{R}$ at each point $x \in \mathcal{X}$ satisfying $\mathrm{E}_x f = f(x)$ is continuous and thus, by the Riesz representation theorem, has a unique Riesz representation, $\eta_{\mathrm{E}_x} \in \mathcal{H}_k$ (Aronszajn, 1943). This Riesz representation $\eta_{\mathrm{E}_x}$ is a partial evaluation of the kernel $k(\cdot, x)$; we refer to it as the kernel embedding of the sample location $x$.

In addition to kernel embeddings of points $k(\cdot, x)$, it is common to study kernel embeddings of measures via Bochner integrals of kernel embeddings or tensor products thereof (Fukumizu et al., 2004). Such work generally approaches $\mathcal{H}_k$ "from the outside"–namely, from $L^2(P)$. Each measure $P$ interacts with the kernel differently, and it is the products of this interaction–eigenvalues and eigenfunctions of the associated integral operator–that are used to characterize $\mathcal{H}_k$.

Suppose we are given a measurable positive-definite kernel $k : \mathcal{X} \times \mathcal{X} \to \mathbb{R}$ on a measurable space $(\mathcal{X}, \mathcal{B})$ with separable reproducing kernel Hilbert space $\mathcal{H}_k$ and a probability measure $P$. Since $\mathcal{H}_k$ is separable and $k$ is measurable, $x \mapsto k(\cdot, x) \otimes k(\cdot, x)$ is strongly measurable.

We suppose, moreover, that $x \mapsto k(\cdot, x)$ is Bochner integrable

$$\int_{\mathcal{X}} ||k(\cdot, x)||_{\mathcal{H}_k} \, \mathrm{d}P(x) = \int_{\mathcal{X}} \sqrt{k(x, x)} \, \mathrm{d}P(x) < \infty \text{ (satisfied, for instance, if } k \text{ is bounded)}.$$

Then the kernel mean embedding of $P$ in $\mathcal{H}_k$ is the Bochner integral that averages the embedding of the samples over the distribution:

$$\mu_P \overset{\text{def}}{=} \int_{\mathcal{X}} k(\cdot, x) \, \mathrm{d}P(x). \tag{4}$$

Since $k(\cdot, x)$ reproduces evaluation at $x$ (i.e., $f(x) = \langle f, k(\cdot, x)\rangle_{\mathcal{H}_k}$), the mean embedding reproduces the expectation of a function:

$$\text{for all } f \in \mathcal{H}_k, \; \mathbb{E}_P[f(x)] = \int_{\mathcal{X}} f(x) \, \mathrm{d}P(x) = \left\langle f, \int_{\mathcal{X}} k(\cdot, x) \, \mathrm{d}P(x) \right\rangle_{\mathcal{H}_k} = \langle f, \mu_P \rangle_{\mathcal{H}_k}.$$

A kernel is called characteristic if the mean embedding is injective:

$$\mu_P = \mu_Q \implies P = Q.$$

The Gaussian and Laplacian kernels are characteristic; the linear kernel is not characteristic because

$$\mu_P = \int_{\mathcal{X}} \langle \cdot, x \rangle_{\mathcal{H}_k} \, \mathrm{d}P(x) = \mathbb{E}_p[x]$$

only captures the first moment. Many distinct distributions have the same mean.

For characteristic kernels, the maximum mean discrepancy

$$MMD(P, Q) \overset{\text{def}}{=} ||\mu_P - \mu_Q||_{\mathcal{H}_k}$$

is a metric.

Just as the kernel mean embedding is the mean of the pointwise kernel embeddings $k(\cdot, x)$, the covariance operator gives the variance of the pointwise embeddings. It is the mean of the rank-one operator $k(\cdot, x) \otimes k(\cdot, x)$ rather than $k(\cdot, x)$. The operator $k(\cdot, x) \otimes k(\cdot, x)$ in $\mathrm{HS}(\mathcal{H}_k)$ acts on $\mathcal{H}_k$ as follows:

$$(k(\cdot, x) \otimes k(\cdot, x))f = \langle f, k(\cdot, x)\rangle_{\mathcal{H}_k} k(\cdot, x) = f(x)k(\cdot, x); \text{ thus, } \langle (k(\cdot, x) \otimes k(\cdot, x))f, g\rangle_{\mathcal{H}_k} = f(x)g(x).$$

The space $\mathrm{HS}(\mathcal{H}_k)$ of Hilbert-Schmidt operators on $\mathcal{H}_k$ is a Hilbert space with norm $||\mathrm{A}||_{\mathrm{HS}}^2 = \sum_{i=0}^{\infty} ||\mathrm{A}\phi_i||_{\mathcal{H}_k}^2$ where $\{\phi_i\}_{i=0}^{\infty}$ is any complete orthonormal system for $\mathcal{H}_k$. Since $k(\cdot, x) \otimes k(\cdot, x)$ is a rank-one operator, the Hilbert-Schmidt norm is simply $||k(\cdot, x) \otimes k(\cdot, x)||_{\mathrm{HS}}^2 = ||k(\cdot, x)||_{\mathcal{H}_k}^2 \cdot ||k(\cdot, x)||_{\mathcal{H}_k}^2 = k(x, x)^2$.

We suppose that the map $x \mapsto k(\cdot, x) \otimes k(\cdot, x)$ is Bochner integrable

$$\int_{\mathcal{X}} ||k(\cdot, x) \otimes k(\cdot, x)||_{\mathrm{HS}} \, \mathrm{d}P(x) = \int_{\mathcal{X}} k(x, x) \, \mathrm{d}P(x) < \infty \text{ (satisfied when } k \text{ is bounded)}$$

and define the uncentered covariance operator $\Sigma_P : \mathcal{H}_k \to \mathcal{H}_k$ of the random variable $k(\cdot, x)$ for $x \sim P$:

$$\Sigma_P = \int_{\mathcal{X}} k(\cdot, x) \otimes k(\cdot, x) \, \mathrm{d}P(x).$$

For all $f$ and $g$ in $\mathcal{H}_k$,

$$\mathbb{E}_P[f(x)g(x)] = \int_{\mathcal{X}} f(x)g(x) \, \mathrm{d}P(x) = \int_{\mathcal{X}} \langle f, k(\cdot, x) \rangle_{\mathcal{H}_k} g(x) \, \mathrm{d}P(x)$$

$$= \left\langle f, \left( \int_{\mathcal{X}} k(\cdot, x) \otimes k(\cdot, x) \, \mathrm{d}P(x) \right) g \right\rangle_{\mathcal{H}_k} = \langle f, \Sigma_P g \rangle_{\mathcal{H}_k}.$$

Since $k$ is symmetric, $\Sigma_P$ is self-adjoint, and $\Sigma_P$ is positive: $\langle f, \Sigma_P f \rangle_{\mathcal{H}_k} = \int_{\mathcal{X}} f(x)^2 \mathrm{d}P(x) \geq 0$. Since $\mathrm{tr}(k(\cdot, x) \otimes k(\cdot, x)) = ||k(\cdot, x)||^2_{\mathcal{H}_k} = k(x, x)$, $\Sigma_P$ is trace-class and thus Hilbert-Schmidt and compact. By the spectral theorem for compact self-adjoint operators on $L^2(P)$ (or a version of Mercer's theorem stated in sufficient generality), there is a complete orthonormal system $\{\phi_i\}_{i=0}^{\infty}$ for $L^2(P)$ such that

$$k(x, y) = \sum_{i=0}^{\infty} \lambda_i \phi_i(x) \phi_i(y)$$

and $\{\lambda_i\}_{i=0}^{\infty}$ (which are in $\in \ell^1$ since we assumed $\int_{\mathcal{X}} k(x, x) \, \mathrm{d}P(x) < \infty$) are the corresponding eigenvalues of the operator $\iota_P \circ \mathrm{T}_{k,P}$ where $\iota_P$ is the inclusion map $\mathcal{H}_k \hookrightarrow L^2(P)$ and $\mathrm{T}_{k,P}$ is its adjoint:

$$\mathrm{T}_{k,P} : L^2(P) \to \mathcal{H}_k$$

$$f \mapsto \int_{\mathcal{X}} k(\cdot, x) f(x) \, \mathrm{d}P(x)$$

The inclusion map $\iota_P$ is continuous since kernel is assumed bounded; this ensures the adjoint $\mathrm{T}_{k,P}$ is well-defined.

We stress that for each distribution $P$, the operator $\mathrm{T}_{k,P}$ maps $L^2(P)$ to the unique RKHS associated with the kernel, $\mathcal{H}_k$, which is the closure of the span of the embeddings $k(\cdot, x)$ for $x \in \mathcal{X}$. However, the eigenvalues $\lambda_i$ and eigenvectors $\phi_i$ of $\iota_P \circ \mathrm{T}_{k,P}$ depend on the choice of distribution $P$. Setting $P$ to be the uniform measure on the sphere, Wahba and Wendelberger (Wahba, 1981; Wendelberger, 1981) found an expansion for the thin-plate splines on the sphere in eigenfunctions of the inclusion map adjoint $\mathrm{T}_{k,P}$; this is discussed in the following section.

While $\mathrm{T}_{P,k}$ smooths a square integrable function (equivalence class) with the kernel, to assure membership in $\mathcal{H}_k$, before viewing it again as an element of $L^2(P)$, the operator $\Sigma_P : \mathcal{H}_k \to \mathcal{H}_k$ views an element of $\mathcal{H}_k$ (sufficiently well-behaved to be seen as a function) as an element of $L^2(P)$ and then further smooths it.

The eigenvalues of the Hilbert-Schmidt operator $\Sigma_P = \mathrm{T}_{k,P} \circ \iota_P$ are the same as those of $\iota_P \circ \mathrm{T}_{k,P}$. While $\{\phi_i\}_{i=0}^{\infty}$ form a complete orthonormal system for $L^2(P)$, the renormalized eigenvectors $\psi_i = \sqrt{\lambda_i} \mathrm{T}_{k,P} \phi_i$ form a complete orthonormal system for $(\mathrm{null}\, \Sigma_P)^{\perp}$ in $\mathcal{H}_k$:

$$\langle \psi_i, \psi_j \rangle_{\mathcal{H}_k} = \frac{1}{\sqrt{\lambda_i \lambda_j}} \langle \mathrm{T}_{k,P} \phi_i, \mathrm{T}_{k,P} \phi_j \rangle_{\mathcal{H}_k} = \frac{1}{\sqrt{\lambda_i \lambda_j}} \langle \phi_i, (\iota_P \circ \mathrm{T}_{k,P}) \phi_j \rangle_{L^2(P)} = \frac{\lambda_j}{\sqrt{\lambda_i \lambda_j}} \langle \phi_i, \phi_j \rangle_{L^2(P)} = \delta_{ij}.$$

The spectral decomposition of the covariance operator is

$$\Sigma_P = \sum_{i=0}^{\infty} \lambda_i (\psi_i \otimes \psi_i),$$

where $\psi_i \otimes \psi_i \in \mathrm{HS}(\mathcal{H}_k)$.

## B. Synthesis of Smoothing Splines on Compact Manifolds and Time-Data Tensor Product Splines

Smoothing splines were initially constructed using the insight that Green's function of the iterated Laplacian $\Delta^m$ on the line is a positive-definite kernel (assuming the sign convention $\Delta = -\text{div grad}$) (Kimeldorf & Wahba, 1971). In Euclidean space, it is a *conditionally* positive-definite kernel, which only slightly complicates the derivation (Wendland, 2004, Section 8.4). For a compact Riemannian manifold $\Omega$, the construction of smoothing splines is especially easy, thanks to Wahba's kernel synthesis procedure.

A smoothing spline is a function $f : \Omega \to \mathbb{R}$ that, given a dataset $(\{x_j\}_{j=1}^n, \{y_j\}_{j=1}^n)$ where $\{x_j\}_{j=1}^n \subseteq \Omega$ and $\{y_j\}_{j=1}^n \subseteq \mathbb{R}$ and scalar $\eta > 0$, solves the following empirical-risk-minimization problem:

$$f^* = \underset{f \in \mathcal{H}}{\text{argmin}} \frac{1}{n} \sum_{j=1}^n (f(x_j) - y_j)^2 + \eta J_{m,\Omega}(f). \tag{5}$$

Here $\mathcal{H}$ is the RKHS (9), associated with the wiggliness penalty $J_{m,\Omega}(f)$, which generalizes the Dirichlet energy wiggliness penalty[5]

$$J_{1,\Omega}(f) = \int_\Omega \|(\nabla f)(x)\|_2^2 \, dx \,.$$

The scalar $\eta$ negotiates the compromise between the adherence of the smoothing spline $f^*$ to the dataset and its smoothness.

On a compact Riemannian manifold without boundary or with appropriate boundary conditions, by Stokes's theorem,

$$0 = \int_\Omega \text{div}(f\nabla f)(x)\, dx = \int_\Omega \|(\nabla f)(x)\|_2^2\, dx - \int_\Omega (\Delta f)(x) \cdot f(x)\, dx \,;$$

thus,

$$J_{1,\Omega} = \int_\Omega (\Delta f)(x) \cdot f(x)\, dx \,.$$

More generally, we can write, for order $m \in \mathbb{N}_+$, the wiggliness penalty

$$J_{m,\Omega}(f) = \begin{cases} \int_\Omega ((\Delta^{m/2} f)(x))^2\, dx\,, & \text{if } m \text{ is even;} \\ \int_\Omega \|\nabla (\Delta^{(m-1)/2} f)(x)\|_2^2\, dx\,, & \text{otherwise;} \end{cases}$$
$$= \int_\Omega f(x)(\Delta^m f)(x)\, dx \,. \tag{6}$$

On a compact Riemannian manifold, the Laplace-Beltrami eigenfunctions $\{\varphi_i\}_{i=0}^\infty$ form a complete orthonormal system for $L^2(\Omega)$, with corresponding eigenvalues $\{\lambda_i\}_{i=0}^\infty$ (by convention, sorted in non-decreasing order) equal to the Dirichlet energy of the mode

$$\lambda_i = J_{1,\Omega}(\varphi_i) = \int_\Omega \|(\nabla \varphi_i)(x)\|_2^2\, dx \geq 0.$$

Thus, any $f \in L^2(\Omega)$ with Fourier expansion

$$f \overset{L^2(\Omega)}{\sim} \sum_{i=0}^\infty \langle f_i, \varphi_i \rangle_{L^2(\Omega)} \varphi_i = \sum_{i=0}^\infty f_i \varphi_i$$

has wiggliness penalty

$$J_{m,\Omega}(f) = \int_\Omega \left( \sum_{i=0}^\infty f_i \varphi_i \right) \left( \sum_{i=0}^\infty f_i \lambda_i^m \varphi_i \right) dx = \sum_{\substack{i=0 \\ \lambda_i > 0}}^\infty \frac{\langle f_i, \varphi_i \rangle_{L^2(\Omega)}^2}{\lambda_i^{-m}}$$
$$= \sum_{i=1}^\infty \frac{f_i^2}{\lambda_i^{-m}} \,, \tag{7}$$

---

[5]We omit the usual factor of $\frac{1}{2}$ in the continuous case but retain it in the discrete case.

since the only solution to the harmonic equation $\Delta f = 0$ on $\Omega$ is the constant function ($\lambda_i = 0$ if and only if $i = 0$ so $\varphi_0$ is constant, as the $\varphi_i$ are sorted by eigenvalue). By Weyl's law (Jost, 2005, Equation 3.2.24), the sequence $(\lambda_i^{-m})_{i=0}^{\infty}$ grows $\Theta(i^{-2m/d})$, where $d$ is the dimension of the manifold. Thus, when $2m > d$, the sequence $(\lambda_i^{-m})_{i=0}^{\infty} \in \ell^1$.

Provided $2m > d$, we can use Wahba's procedure to synthesize an RKHS $\mathcal{H}_1$ and Mercer kernel on the Fourier side.[6] The smoothing spline inner product

$$\langle f, g \rangle_{\mathcal{H}_1} = \int_{\Omega} f(x)(\Delta g)(x)\, \mathrm{d}x = \sum_{i=1}^{\infty} \frac{\langle f, \varphi_i \rangle_{L^2(\Omega)}^2 \langle g, \varphi_i \rangle_{L^2(\Omega)}}{\lambda_i^{-m}}$$

induces the wiggliness penalty norm (6)

$$J_{m,\Omega}(f) = ||f||_{\mathcal{H}_1}^2 = \int_{\Omega} f(x)(\Delta f)(x)\, \mathrm{d}x = \sum_{i=1}^{\infty} \frac{\langle f, \varphi_i \rangle_{L^2(\Omega)}}{\lambda_i^{-m}}.$$

The RKHS $\mathcal{H}_1$ consists of zero-mean finite-energy signals without too much of that finite energy on the high-wiggliness modes:

$$\mathcal{H}_1 = \left\{ f \in L^2(\Omega) \,\middle|\, ||f||_{\mathcal{H}_1}^2 < \infty \text{ and } \langle f, \varphi_0 \rangle_{L^2(\Omega)} = 0 \right\}. \tag{8}$$

The reproducing kernel $k^1$ corresponding to $\mathcal{H}_1$ can be computed using an infinite series:

$$\forall (x, y) \in \Omega^2, k^1(x, y) = \sum_{i=1}^{\infty} \lambda_i^{-m} \varphi_i(x) \varphi_i(y).$$

While the family $\{\varphi_i\}_{i=1}^{\infty}$ is not locally bounded when $d > 1$, the Sogge bound

$$||\varphi_i||_{L^{\infty}(\Omega)} \leq C \lambda_i^{\frac{d-1}{4}} ||\varphi_i||_{L^2(\Omega)}$$

allows us to find $m$ such that this sum converges uniformly for any given dimension (see, e.g., (Burq & Lebeau, 2014), Equations (1.1)-(1.2); (Sogge, 1988), Equation (4.2); or (Zelditch, 2017), Equation (3.3)). Indeed, with $d = 2$, $m = 2$ suffices, as

$$\sum_{i=1}^{\infty} \lambda_i^{-m} \varphi_i(x) \varphi_i(y) \leq C_2 \sum_{i=1}^{\infty} (i^{-\frac{2m}{d}})(i^{\frac{d-1}{4}})(i^{\frac{d-1}{4}}) = C_2 \sum_{i=1}^{\infty} i^{-\frac{3}{2}} < \infty.$$

We can eliminate the zero-mean constraint of (8) via the Decomposition Principle (see Berlinet & Thomas-Agnan, 2011, as well as Remark 7). As the only finite-energy functions in $L^2(\Omega)$ in the null space of the wiggliness penalty $J_{m,\Omega}$ are the constant functions $\mathcal{H}_0 = \mathrm{span}\{\varphi_0\}$, we can incorporate nonzero mean values via the direct sum:

$$\mathcal{H} = \mathcal{H}_0 \oplus \mathcal{H}_1 \tag{9}$$

The one-dimensional space of constant functions $\mathcal{H}_0$ is an RKHS with inner product defined, for any choice of $\overline{\lambda}_0 > 0$, as follows:

$$\langle f, g \rangle_{\mathcal{H}_0} = \overline{\lambda}_0 \langle f, \varphi_0 \rangle_{L^2(\Omega)} \cdot \langle g, \varphi_0 \rangle_{L^2(\Omega)} = \frac{\overline{\lambda}_0}{\mathrm{vol}\Omega} \mathrm{mean}(f) \cdot \mathrm{mean}(g).$$

The Riesz representation of evaluation at any point $x \in \Omega$ for this space of constant functions $\mathcal{H}_0$ is $k_x^0 = (\overline{\lambda}_0 \sqrt{\mathrm{vol}\Omega})^{-1} \varphi_0$ so that

$$f(x) = \mathrm{mean}(f) = \frac{1}{\sqrt{\mathrm{vol}\Omega}} \langle f, \varphi_0 \rangle_{L^2(\Omega)} = \langle f, k_x^0 \rangle_{\mathcal{H}_0}.$$

The spaces $\mathcal{H}_0$ of zero-wiggliness functions with arbitrary mean and $\mathcal{H}_1$ of zero-mean functions with arbitrary nonnegative wiggliness are indeed orthogonal with respect to the inner product

$$\langle f, g \rangle_{\mathcal{H}} = \langle f, g \rangle_{\mathcal{H}_0} + \langle f, g \rangle_{\mathcal{H}_1}$$

$$= \frac{\langle f, \varphi_0 \rangle_{L^2(\Omega)} \langle g, \varphi_0 \rangle_{L^2(\Omega)}}{\overline{\lambda}_0^{-1}} + \sum_{i=1}^{\infty} \frac{\langle f, \varphi_i \rangle_{L^2(\Omega)} \langle g, \varphi_i \rangle_{L^2(\Omega)}}{\lambda_i^{-m}}.$$

---

[6]Note that this space $\mathcal{H}_1$ is not the Sobolev space $H^{1,2}$, often called $\mathcal{H}^1$, but the homogeneous Sobolev space (or Beppo Levi space).

Any $f \in \mathcal{H}$ can be evaluated at any $x \in \Omega$ via the reproducing property using the Riesz representation of evaluation at $x$ in $\mathcal{H}$, $k_x = k_x^0 + k_x^1$:

$$f(x) = \text{mean}(f) + \langle f, k_x^1 \rangle_{\mathcal{H}_1} = \langle f, k_x^0 \rangle_{\mathcal{H}_0} + \langle f, k_x^1 \rangle_{\mathcal{H}_1}$$

$$= \langle f, k_x^0 + k_x^1 \rangle_{\mathcal{H}} = \left\langle f, \frac{1}{\lambda_0\sqrt{\text{vol}\Omega}}\varphi_0 + \sum_{i=1}^{n} \lambda_i^{-m}\varphi_i(x)\varphi_i \right\rangle_{\mathcal{H}}.$$

The Wahba-Kimeldorf representer theorem (Kimeldorf & Wahba, 1971; Schölkopf et al., 2001) guarantees that the solution to (5) over all of $\mathcal{H}$ lies in $\text{span}\{\varphi_0, k_{x_1}^1, \ldots, k_{x_n}^1\}$, where the Riesz representation of evaluation at $x_j$, $k_{x_j}^1 = k^1(\cdot, x_j)$. This is shown via a simple orthogonality argument: the orthogonal complement of the finite-dimensional space $\text{span}\{k_{x_1}^1, \ldots, k_{x_n}^1\}$ in the infinite-dimensional space $\mathcal{H}_1$ cannot affect the data-adherence penalty

$$\text{adherence}(f) = \frac{1}{n}\sum_{j=1}^{n}(f(x_j) - y_j)^2 = \frac{1}{n}\sum_{j=1}^{n}(\langle f, k_{x_j} \rangle_{\mathcal{H}} - y_j)^2$$

because, for $j = 1, \ldots, n$, $f(x_j) = \langle f, k_{x_j}^1 \rangle_{\mathcal{H}_1} = 0$. But the orthogonal projection of $f^*$ onto this orthogonal complement of the Riesz representations of evaluation on the data set adds to the wiggliness penalty! When the solution is of the form

$$f^* = \alpha_0\varphi_0 + \sum_{j=1}^{n}\alpha_j k_{x_j}^1,$$

the data-adherence and wiggliness penalties may be written using matrix algebra; the weights $\alpha_0, \alpha_1, \ldots, \alpha_n$ that locate $f^*$ in $\mathcal{H}$ can be found using an inversion (or, if the $k_{x_j}$ are linearly dependent, a Moore-Penrose pseudo-inverse) of the $n \times n$ Gram matrix of $k$ and a matrix multiplication.

This approach can be extended to case where samples take values in some RKHS $\mathcal{D}$. For example, if $\mathcal{X} = [0, 1]$, let $\mathcal{H}^1$ be the usual sobolev space $W^{2,2}([0, 1])$. Then we can define a the product kernel $k((t, y), (t', y')) = R(t, t')k(y, y')$ on $\mathcal{H}^1 \otimes \mathcal{D}$. We seek to minimize

$$\sum_{i=1}^{n}||f(t_i) - k_{\mathcal{D}}(\cdot, y_i)||_{\mathcal{D}}^2 + \lambda\int_0^1 ||f''(t)||_{\mathcal{D}}^2 \, \mathrm{d}t$$

where $f''$ is the second-order Bochner derivative. The solution, by the representer theorem, is

$$f^* = \sum_{i=1}^{n}\alpha_i(t)k_{\mathcal{D}}(\cdot, y_i)$$

where $\alpha_i$ are found by solving $\alpha_i(t)^T = r(t)^T\mathbf{R}^{-1}$ and $\mathbf{R}_{i,j} = R(t_i, t_j)$ is the $n \times n$ kernel Gram matrix and $r(t) = [R(t, t_1), \ldots, R(t, t_n)]$.

## C. Synthesis of Kernels on the Fourier Side

### C.1. Wahba's Synthesis of Mercer Kernels on Compact Riemannian Manifolds

Wahba and Wendelberger (Wahba, 1981; 1990; Wendelberger, 1981) developed a method to synthesize "on the Fourier side" continuous positive-definite kernels on a compact Riemannian manifold $\Omega$ (Dunitz, 2023, Proposition 2.38).[7] Given a complete orthonormal system $\{\varphi_i\}_{i=0}^{\infty}$ for $L^2(\Omega)$, one can synthesize a continuous kernel given a nonnegative sequence $(\lambda_i)_{i=0}^{\infty}$ in $\ell^1$:

$$\forall(x, y) \in \Omega^2, k(x, y) = \sum_{i=0}^{\infty}\lambda_i\varphi_i(x)\varphi_i(y). \tag{10}$$

If $\Omega$ is compact, each $\varphi_i$ continuous, and uniform convergence of the Mercer expansion (10) is assured (e.g., via local boundedness of the family $\{\varphi_i\}_{i=0}^{\infty}$ or via Sobolev embedding theorems), then $k$ is continuous and its corresponding reproducing kernel Hilbert space $\mathcal{H}$

---

[7]While a great deal of prior work in spline theory, such as Schoenberg's work on cardinal splines and the fundamental spline, echoes this approach, it was not until the early 1970s that it was used–by Schoenberg (Schoenberg, 1972) and v. Golitschek (v. Golitschek, 1972)–to synthesize smoothing splines on a compact manifold (the circle).

1. has inner product

$$\langle f, g \rangle_{\mathcal{H}} = \sum_{\substack{i=0 \\ \lambda_i > 0}}^{\infty} \frac{\langle f, \varphi_i \rangle_{L^2(\Omega)} \langle g, \varphi_i \rangle_{L^2(\Omega)}}{\lambda_i};$$

2. consists of *continuous* functions $f : \Omega \to \mathbb{R}$ of finite norm with respect to the norm induced by the inner product and (for definiteness) no energy on the modes associated with $\lambda_i = 0$:

$$||f||_{\mathcal{H}}^2 = \sum_{\substack{i=0 \\ \lambda_i > 0}}^{\infty} \frac{\langle f, \varphi_i \rangle_{L^2(\Omega)}}{\lambda_i} < \infty \text{ and } \sum_{\{i : \lambda_i = 0\}} \langle f, \varphi_i \rangle_{L^2(\Omega)}^2 = 0;$$

3. has a complete orthonormal system $\{\sqrt{\lambda_i} \varphi_i \,|\, \lambda_i > 0\}_{i=0}^{\infty}$ constructed from eigenvectors $\varphi_i$ and eigenvalues $\lambda_i$ of the Hilbert-Schmidt integral operator[89]

$$\mathrm{L}_k : L^2(\Omega) \to L^2(\Omega)$$

$$f \overset{L^2(\Omega)}{=} \sum_{i=0}^{\infty} f_i \varphi_i \mapsto \int_{\Omega} k(\cdot, y) f(y) \, \mathrm{d}y = \sum_{i=0}^{\infty} \lambda_i f_i \varphi_i$$

associated with the kernel $k$;

4. has the following embedding $\Phi$ from $x \in \Omega$ to the Riesz representation $k_x \in \mathcal{H}$ of the evaluation functional at $x$:

$$\Phi(x) = k_x = \sum_{i=0}^{\infty} \lambda_i \varphi_i(x) \varphi_i; \text{ and}$$

5. is the image of $L^2(\Omega)$ under the isometry $\mathrm{L}_k^{1/2}$

$$\mathrm{L}_k^{1/2} : L^2(\Omega) \to L^2(\Omega)$$

$$f \overset{L^2(\Omega)}{=} \sum_{i=0}^{\infty} f_i \varphi_i \mapsto \sum_{i=0}^{\infty} \sqrt{\lambda_i} f_i \varphi_i$$

since

$$\left\| \mathrm{L}_k^{1/2} f \right\|_{\mathcal{H}} = \sum_{i=0}^{\infty} \frac{(\sqrt{\lambda_i} f_i)^2}{\lambda_i} = ||f||_{L^2(\Omega)}.$$

### C.2. Wahba's Synthesis on Discrete Domains

Suppose we are given an index set $\mathcal{X} = \{1, \ldots, T\}$, a diagonal matrix with nonnegative entries $\Lambda = \mathrm{diag}(\lambda_1, \ldots, \lambda_T) \succeq 0$, and a $T \times T$ orthogonal matrix $\mathbf{U}$, whose columns $u_1, \ldots, u_T$ form an orthonormal basis of $\mathbb{R}^T$. With $\mathcal{X}$ playing the part of $\Omega$, $\mathbb{R}^T$ the part of $L^2(\Omega)$, $\Lambda$ of $\{\lambda_i\}_{i=0}^{\infty}$, $\mathbf{U}$ of $\{\varphi_i\}_{i=0}^{\infty}$, we can establish analogous results. Let $\mathbf{K} = \mathbf{U} \Lambda \mathbf{U}^T$ and $e_i$ be the $i$th standard basis function of $\mathbb{R}^T$.

The kernel $k : \mathcal{X}^2 \to \mathbb{R}$ on our graph or discrete domain, defined by

$$\forall (i, j) \in \mathcal{X}^2, k(i, j) = \sum_{t=1}^{T} \lambda_t u_t(i) u_t(j) = e_i^T \mathbf{U} \Lambda \mathbf{U}^T e_j = (\mathbf{K})_{i,j},$$

---

[8]Crucially, since $k$ is continuous and $\Omega$ compact, we have

$$\iint\limits_{\Omega \times \Omega} |k(x, y)|^2 \, \mathrm{d}x \, \mathrm{d}y = B < \infty.$$

[9]Using the notation of the previous section $\mathrm{L}_k = \iota_P \circ \mathrm{T}_{k,P}$ where $P$ is the uniform measure on $\Omega$.

has as Gram matrix $\mathbf{K} = \mathbf{U}\boldsymbol{\Lambda}\mathbf{U}^T$ (computed over the entire discrete space $\mathcal{X} = \{1, \ldots, T\}$).

The corresponding RKHS $\mathcal{H} \subseteq \mathbb{R}^{\{1,\ldots,T\}}$ (which we identify with $\mathbb{R}^T$) is $\text{span}\{u_i \mid \lambda_i > 0\}_{i=1}^T$ is a space of *graph functions* or functions on the discrete domain.

1. has inner product

$$\langle f, g \rangle_{\mathcal{H}} = \sum_{\substack{i=1 \\ \lambda_i > 0}}^{T} \frac{\langle f, u_i \rangle_{\mathbb{R}^T} \langle g, u_i \rangle_{\mathbb{R}^T}}{\lambda_i} = f^T \mathbf{K}^\dagger g = \langle \mathbf{W}f, \mathbf{W}g \rangle_{\mathbb{R}^T},$$

   where $\mathbf{W} = (\boldsymbol{\Sigma}^\dagger)^{1/2}\mathbf{U}^T$ satisfies $\mathbf{W}^T\mathbf{W} = \mathbf{K}^\dagger$;

2. has complete orthonormal system $\{\sqrt{\lambda_i}u_i \mid \lambda_i > 0\}_{i=1}^T$ constructed from eigenvectors $u_i$ and corresponding eigenvalues $\lambda_i$ of the operator

$$\mathrm{L}_k : \mathbb{R}^T \to \mathbb{R}^T$$

$$f \mapsto \mathbf{K}f = \mathbf{U}\boldsymbol{\Lambda}\mathbf{U}^T f = \sum_{\substack{i=1 \\ \lambda_i > 0}}^{T} \lambda_i \langle f, u_i \rangle_{\mathbb{R}^T} u_i;$$

3. is the image of $\mathbb{R}^T$ under $\mathrm{L}_k$ or under the isometry $\mathrm{L}_k^{1/2}$

$$\mathrm{L}_k^{1/2} : \mathbb{R}^T \to \mathbb{R}^T$$

$$f = \sum_{i=1}^{T} \langle f, u_i \rangle_{\mathbb{R}^T} u_i \mapsto \sum_{i=1}^{T} \sqrt{\lambda_i} \langle f, u_i \rangle_{\mathbb{R}^T} u_i = \mathbf{U}\boldsymbol{\Lambda}^{1/2}\mathbf{U}^T f$$

   as

$$\left\| \mathrm{L}_k^{1/2} f \right\|_{\mathcal{H}} = \sum_{\substack{i=1 \\ \lambda_i > 0}}^{T} \frac{\lambda_i \langle f, u_i \rangle_{\mathcal{H}}^2}{\lambda_i} = \sum_{i=1}^{T} \langle f, u_i \rangle_{\mathbb{R}^T}^2 = \|f\|_{\mathbb{R}^T}^2;$$

4. has the following embedding $\Phi$ from $t \in \{1, \ldots, T\}$ to the Riesz representation $k_t$:

$$\Phi : \{1, \ldots, T\} \to \mathcal{H}$$

$$t \mapsto \Phi(t) = k_t = \sum_{i=1}^{T} \lambda_i u_i(t) u_i,$$

   where $u_i(t) = \langle e_t, u_i \rangle_{\mathbb{R}^T}$ is the $t$th element of $u_i$. Indeed, for any $f \in \text{span}\{u_i \mid \lambda_i > 0\}_{i=1}^T = \mathcal{H}$, we can verify, for all $t \in \{1, \ldots, T\}$, that the reproducing property holds:

$$\langle f, k_t \rangle_{\mathcal{H}} = \sum_{\substack{i=1 \\ \lambda_i > 0}}^{T} \frac{\langle f, u_i \rangle_{\mathbb{R}^T} \left\langle \sum_{i=1}^{T} \lambda_i u_i(t) u_i, u_i \right\rangle_{\mathbb{R}^T}}{\lambda_i}$$

$$= \sum_{\substack{i=1 \\ \lambda_i > 0}}^{T} u_i(t) \langle f, u_i \rangle_{\mathbb{R}^T} = \left\langle e_i, \sum_{\substack{i=1 \\ \lambda_i > 0}}^{T} \langle f, u_i \rangle_{\mathbb{R}^T} u_i \right\rangle_{\mathbb{R}^T} = f(t).$$

**Remark 5.** *The numerator terms can be "Fourier expansions" in a more literal sense. When $\mathbf{L}$ is the graph Laplace matrix, $\mathbf{U}^T$ performs what is often called the graph Fourier transform. For chain graphs it corresponds to the DCT-II and for cyclic graphs the DFT.*

**Remark 6.** *The transformation $\mathbf{W}$ in part (1) can be seen as a* whitening filter*: if $\text{cov}(x, x) = \mathbf{K} = \mathbf{U}\boldsymbol{\Lambda}\mathbf{U}^T$, then*

$$\text{cov}(\mathbf{W}x, \mathbf{W}x) = \mathbf{W}\mathbb{E}[xx^T]\mathbf{W}^T = (\boldsymbol{\Lambda}^\dagger)^{1/2}\mathbf{U}^T\mathbf{U}\boldsymbol{\Lambda}\mathbf{U}^T\mathbf{U}(\boldsymbol{\Lambda}^\dagger)^{1/2} = \mathbf{I}.$$

As $\mathcal{H} = \text{span}\{u_i \mid \lambda_i > 0\}_{i=1}^T$, if there are any $i$ for which $\lambda_i = 0$, then $\mathcal{H}$ is a proper subspace of $\mathbb{R}^T$. The inner product $\langle \cdot, \cdot \rangle_{\mathcal{H}}$ is definite only over $\mathcal{H}$; it is indefinite over $\mathbb{R}^T$. If we wish to make definite, over $\mathbb{R}^T$, the semi-inner product $\langle \cdot, \cdot \rangle_{\mathcal{H}}$, we must define an inner product over the component of any vector $v \in \mathbb{R}^T$ that is not in $\mathcal{H}$. For kernels on graphs or other discrete sets, this is useful for applying smoothness penalties for interpolation applications (e.g., we might not wish to subject any constant component $u_1 = 1_T$–or for disconnected graphs, other Laplace matrix eigenvectors associated with eigenvalue 0–of the interpolant to a smoothness penalty). When using kernels to compare graph signals or other functions on $\mathbb{R}^T$, we may wish to make $\langle \cdot, \cdot \rangle_{\mathcal{H}}$ definite to ensure that modes $u_i$ associated with eigenvalue 0 factor into the comparison. In both situations, we can reinforce $\langle \cdot, \cdot \rangle_{\mathcal{H}}$ by defining an inner product on its orthogonal complement in $\mathbb{R}^T$. In effect, we swap out all $\lambda_i = 0$ with some $p_i > 0$.

**Remark 7** (Decomposition Principle). *Suppose we are given an RKHS $\mathcal{H} = \text{span}\{u_i \mid \lambda_i > 0\}_{i=1}^T$ with inner product $\langle \cdot, \cdot \rangle_{\mathcal{H}}$. We can define the RKHS $\mathcal{H}_0 = \text{span}\{u_i \mid \lambda_i = 0\}_{i=1}^T$ with inner product*

$$\langle f, g \rangle_{\mathcal{H}_0} = \sum_{\substack{i=1 \\ \lambda_i = 0}}^T \frac{\langle f, u_i \rangle_{\mathbb{R}^T} \langle g, u_i \rangle_{\mathbb{R}^T}}{p_i},$$

*where the $p_i$ may be any nonnegative number (but are often arbitrarily set to 1). This decomposition was introduced by Wahba and Kimeldorf in the context of smoothing splines (Kimeldorf & Wahba, 1971). In this setting, we wish to apply a wiggliness penalty on $\mathcal{H}$ and exempt the orthogonal projection $\mathbf{P}_0 f = \mathbf{U}_0 \mathbf{U}^T$ of any $f \in \mathbb{R}^T$ onto $\text{span}\{u_i \mid \lambda_i = 0\}_{i=1}^T$ (where $\mathbf{U}_0$ is the semi-orthogonal matrix with columns $\{u_i \mid \lambda_i = 0\}_{i=1}^T$). (See (Berlinet & Thomas-Agnan, 2011; Kimeldorf & Wahba, 1971; Wahba, 1990).) However, it is of great use in the context of graph function kernels: we may wish to preserve the DC component in comparing two signals defined on the vertices of our graph, though kernels based on the graph Laplace matrix can only effectively compare zero-mean signals.*

## D. FFT-Implementation for Smooth, Axis-Aligned Gridded Data

When $G = (V, E)$ is an unweighted chain graph or a cycle graph (see Fig. 7) and $f \in \mathbb{R}^T$, the product $\mathbf{U}^T f = \widehat{f}$ and the Dirichlet energy $f^T \mathbf{L} f = \sum_{i=1}^T \lambda_i \widehat{f}(i)^2$ can each be computed in $\Theta(|V| \log(|V|)) = \Theta(T \log T)$ time using the FFT implementation of the discrete cosine transform (DCT-II) or discrete Fourier transform (DFT), respectively. When we instead have vector-valued observations $y_i \in \mathcal{Y}$ at each vertex, the Dirichlet energy can be computed in $O(T^2 \log T)$ time, provided we have access to the Gram matrix $\mathbf{G}$, no matter the dimension of $\mathcal{Y}$.

Thanks to the Kronecker product, this trick is especially useful with high-dimensional gridded data. For instance, let $\mathbf{L}_{\text{chain}} = \mathbf{P}\widetilde{\mathbf{\Lambda}}\mathbf{P}^T$ be the Laplace matrix of a length-$n$ chain (representing a row or column of an image). Then the Laplace matrix of the $n \times n$ image graph $\mathbf{L}_{\text{image}} = \mathbf{L}_{\text{chain}} \otimes \mathbf{I}_{n \times n} + \mathbf{I}_{n \times n} \otimes \mathbf{L}_{\text{chain}}$. Notice that, by the distributive and mixed-product properties of the Kronecker product,

$$\begin{aligned}
\mathbf{L}_{\text{image}}(\rho_i \otimes \rho_j) &= (\mathbf{L}_{\text{chain}} \otimes \mathbf{I}_{n \times n} + \mathbf{I}_{n \times n} \otimes \mathbf{L}_{\text{chain}})(\rho_i \otimes \rho_j) \\
&= (\mathbf{L}_{\text{chain}} \otimes \mathbf{I}_{n \times n})(\rho_i \otimes \rho_j) + \\
&\quad (\mathbf{I}_{n \times n} \otimes \mathbf{L}_{\text{chain}})(\rho_i \otimes \rho_j) \\
&= (\mathbf{L}_{\text{chain}}\rho_i) \otimes (\mathbf{I}_{n \times n}\rho_j) + (\mathbf{I}_{n \times n}\rho_i) \otimes (\mathbf{L}_{\text{chain}}\rho_j) \\
&= \widetilde{\lambda}_i \rho_i \otimes \rho_j + \widetilde{\lambda}_j \rho_i \otimes \rho_j = (\widetilde{\lambda}_i + \widetilde{\lambda}_j)\rho_i \otimes \rho_j,
\end{aligned}$$

where $\rho_i$ are the columns of $\mathbf{P}$ and $\widetilde{\lambda}_i$ the diagonal of the diagonal matrix $\widetilde{\mathbf{\Lambda}}$. Thus, the eigenvectors of $\mathbf{L}_{\text{image}}$ are $\rho_i \otimes \rho_j$ with corresponding eigenvalues $\widetilde{\lambda}_i + \widetilde{\lambda}_j$. The modes of a chain graph and their Dirichlet energies are presented in Figure 7. From this perspective, we can see why the kernel $\mathbf{L}^\dagger$ is natural: for a chain graph, the Laplace matrix $\mathbf{L}_{\text{chain}}$ looks like the precision matrix of a stationary $AR(1)$ process $y_t = \alpha y_{t-1} + \varepsilon_t$, with $|\alpha| < 1$ and $\varepsilon_t \sim \mathcal{N}(0, \sigma^2)$. This precision matrix $\frac{1}{\sigma^2}\mathbf{C}^{-1}$ is tridiagonal and Toeplitz with $(\mathbf{C}^{-1})_{1,1} = (\mathbf{C}^{-1})_{n,n} = 1$ and otherwise $1 + \alpha^2$ on the diagonal and $-\alpha$ just off

the diagonal. Indeed, when $\alpha = 1 - \epsilon$, we obtain

$$\mathbf{C}^{-1} = \begin{pmatrix} 1 & -1+\epsilon & 0 & \ldots & 0 & 0 \\ -1+\epsilon & 2-2\epsilon+\epsilon^2 & -1+\epsilon & \ldots & 0 & 0 \\ 0 & -1+\epsilon & 2-2\epsilon+\epsilon^2 & \ldots & 0 & 0 \\ \vdots & \vdots & \vdots & \ddots & \vdots & \vdots \\ 0 & 0 & 0 & \ldots & 2-2\epsilon+\epsilon^2 & -1+\epsilon \\ 0 & 0 & 0 & \ldots & -1+\epsilon & 1 \end{pmatrix}.$$

Thus, $\mathbf{C}^{-1}$ (invertible for $|\alpha| < 1$) tends toward $\mathbf{L}_{\text{chain}}$ (singular) in, for instance, the Frobenius norm, as $\alpha \to 1^-$. From i.i.d. stationarity, we get that $\gamma(0) = \mathbf{C}_{i,i} = \mathbb{E}[y_i^2] = \alpha\gamma(1) + \sigma^2$; for $\tau > 0$, $\gamma(\tau) = \alpha\gamma(\tau-1)$, where $\gamma(\tau) = \mathbb{E}[y_t y_{t+\tau}]$. From this recursive relationship, we identify the covariance matrix $\mathbf{C} = \mathbb{E}[y_i y_j] = \frac{\sigma^2}{1-\alpha^2}\alpha^{|i-j|}$ for $|\alpha| < 1$. This connection was widely noted during the development of JPEG 1 standard. The singularity at $\alpha = 1$ matters a great deal, however, as $\mathbf{L}_{\text{chain}}^T$ assumes a substantially more complicated form than $\mathbf{C}$ and resembles the continuous Green's function for Poisson's equation. In particular,

$$(\mathbf{L}_{\text{chain}}^{\dagger})_{i,j} = \frac{1}{T+1}\left(\min(i,j) \cdot (T+1-\max(i,j))\right).$$

Thus, we can efficiently compute graph kernels on gridded data. First, assemble the graph Laplacian from the eigenvalues and eigenvectors of each component dimension of the grid. The grid dimensions can be different. Certain dimensions can be periodic (cycles) and others chains. Combining periodic and non-periodic axes can be useful for time series with seasonal variation but multiyear trends, InSAR data, cylindrical borehole data, etc.). The eigenvalues of a component dimension's graph Laplace matrix can be scaled, which corresponds to the application of a weight to those edges. All we need is a grid that is uniform and FFT-implementable along each dimension. Next, we can compute a graph kernel $k(f,g) = \text{vec}(f)^T h(\mathbf{L}_{\text{grid}})\text{vec}(g)$ (linear kernel on gridded data where $h$ governs our choice of kernel (heat, Sobolev, iterated Laplacian, etc.) on the grid) or nonlinear kernels such as $k(f,g) = (\text{vec}(f)^T h(\mathbf{L}_{\text{grid}})\text{vec}(g) + d)^m$ (polynomial), $k(f,g;\sigma) = \exp\left(-\frac{1}{\sigma^2}\text{vec}(f-g)^T h(\mathbf{L}_{\text{grid}})\text{vec}(f-g)\right)$ (Gaussian), and so forth.

## E. We Can Plug Temporally Aware Kernels into Traditional Kernel Change-Point Detection Methods

In the traditional RKHS change-point detection setting, $\mathcal{X} = \mathbb{R}^d$ is an index set of time series *values* (not sample locations). We choose a characteristic positive-definite kernel $k : \mathcal{X} \times \mathcal{X} \to \mathbb{R}$ and denote as $\mathcal{H}$ its RKHS.

The kernel mean embedding of of a measure $p$

$$\mu_k(p) = \int_{\mathcal{X}} k(\cdot, x)\mathrm{d}p(x)$$

has empirical estimate over a bag of samples $\mathcal{D} \subseteq \mathcal{X}$:

$$\widehat{\mu}_k(\mathcal{D}) = \frac{1}{|\mathcal{D}|}\sum_{x \in \mathcal{D}} k(\cdot, x).$$

Thus, for any $f \in \mathcal{H}$, the empirical mean of $f$ on the dataset $\mathcal{D}$ can be found via an inner product between $f$ and $\widehat{\mu}_k(\mathcal{D})$:

$$\frac{1}{|\mathcal{D}|}\sum_{x \in \mathcal{D}} f(x) = \frac{1}{|\mathcal{D}|}\sum_{x \in \mathcal{D}}\langle f, k(\cdot, x)\rangle_{\mathcal{H}} = \left\langle f, \frac{1}{|\mathcal{D}|}\sum_{x \in \mathcal{D}} k(\cdot, x)\right\rangle_{\mathcal{H}} = \langle f, \widehat{\mu}_k(\mathcal{D})\rangle_{\mathcal{H}}.$$

We express the usual sample variance of a set $\mathcal{D}$ of time series samples using the canonical dual space isometry

$$\text{var}(\mathcal{D}) = \frac{1}{|\mathcal{D}|}\sum_{x \in \mathcal{D}}(x-\mu)^T(x-\mu) = \frac{1}{|\mathcal{D}|}\sum_{x \in \mathcal{D}}||x-\mu||_{\mathbb{R}^d}^2 = \frac{1}{|\mathcal{D}|}\sum_{x \in \mathcal{D}}||\langle \cdot, x\rangle_{\mathbb{R}^d} - \mu^*||_{(\mathbb{R}^d)^*}^2,$$

with empirical mean vector and covector

$$\mu = \frac{1}{n}\sum_{x \in \mathcal{D}} x \text{ and } \mu^* = \langle \cdot, \mu\rangle_{\mathbb{R}^d} = \frac{1}{n}\sum_{x \in \mathcal{D}}\langle \cdot, x\rangle_{\mathbb{R}^d}.$$

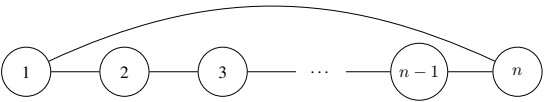

*(a)* A chain graph. Its automorphisms: reflection about center.

| Laplace Matrix | Spectral Decomposition |
|---|---|
| $\mathbf{L}_{\text{chain}} = \begin{pmatrix} 1 & -1 & 0 & \ldots & 0 & 0 \\ -1 & 2 & -1 & \ldots & 0 & 0 \\ 0 & -1 & 2 & \ldots & 0 & 0 \\ \vdots & \vdots & \vdots & \ddots & \vdots & \vdots \\ 0 & 0 & 0 & \ldots & 2 & -1 \\ 0 & 0 & 0 & \ldots & -1 & 1 \end{pmatrix}$ | $\mathbf{L}_{\text{chain}} = \mathbf{U}_{\text{chain}} \text{diag}(\lambda_1, \ldots, \lambda_n) \mathbf{U}_{\text{chain}}^T,$ where $\lambda_i = 2 - 2\cos\left(\frac{\pi(i-1)}{n}\right)$ $(\mathbf{U}_{\text{chain}})_{j,k} = \begin{cases} \frac{1}{\sqrt{n}}, & \text{if } k = 1; \\ \sqrt{\frac{2}{n}}\cos\left(\frac{\pi(k-1)\left(j-\frac{1}{2}\right)}{n}\right), & \text{if } k > 1 \end{cases}$ |

*(b)* The DCT-2 basis functions diagonalize its Laplace matrix.

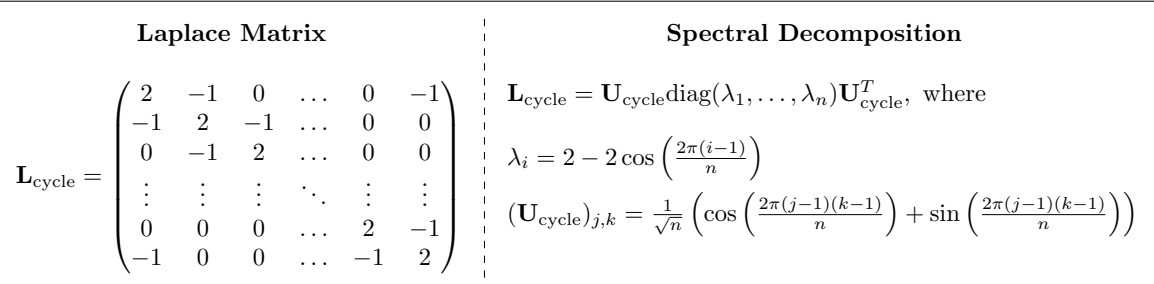

*(c)* A 1-torus. Its automorphisms: cyclic shifts and reflections.

| Laplace Matrix | Spectral Decomposition |
|---|---|
| $\mathbf{L}_{\text{cycle}} = \begin{pmatrix} 2 & -1 & 0 & \ldots & 0 & -1 \\ -1 & 2 & -1 & \ldots & 0 & 0 \\ 0 & -1 & 2 & \ldots & 0 & 0 \\ \vdots & \vdots & \vdots & \ddots & \vdots & \vdots \\ 0 & 0 & 0 & \ldots & 2 & -1 \\ -1 & 0 & 0 & \ldots & -1 & 2 \end{pmatrix}$ | $\mathbf{L}_{\text{cycle}} = \mathbf{U}_{\text{cycle}} \text{diag}(\lambda_1, \ldots, \lambda_n) \mathbf{U}_{\text{cycle}}^T,$ where $\lambda_i = 2 - 2\cos\left(\frac{2\pi(i-1)}{n}\right)$ $(\mathbf{U}_{\text{cycle}})_{j,k} = \frac{1}{\sqrt{n}}\left(\cos\left(\frac{2\pi(j-1)(k-1)}{n}\right) + \sin\left(\frac{2\pi(j-1)(k-1)}{n}\right)\right)$ |

*(d)* The unitary DFT matrix $\mathbf{D}_n$ diagonalizes its Laplace matrix. We prefer the real eigenvectors $\mathbf{U}_{\text{cycle}} = \mathfrak{Re}\{\mathbf{D}_n\} - \mathfrak{Im}\{\mathbf{D}_n\}$.

*Figure 7.* Common FFT-implementable kernels are based on unweighted graphs. One-dimensional chain and cycle graphs–and their spectra. Other boundary conditions, involving edges of weight 2 (treated in Strang, 1999), correspond to DCT-1, DCT-5, and DCT-6.

Notice that when $k$ is the linear kernel $k(x, y) = \langle x, y \rangle_{\mathbb{R}^d}$, then $\mathcal{H} = (\mathbb{R}^d)^*$ the kernel embedding of a point $\phi(x) = k(\cdot, x) = \langle \cdot, x \rangle_{\mathbb{R}^d}$. We can now write $\text{var}(\mathcal{D})$ in kernel form:

$$\text{var}(\mathcal{D}) = \frac{1}{|\mathcal{D}|}\sum_{x \in \mathcal{D}} ||k(\cdot, x) - \widehat{\mu}(\mathcal{D})||_{\mathcal{H}}^2 \text{ with } \widehat{\mu}(\mathcal{D}) = \frac{1}{|\mathcal{D}|}\sum_{x \in \mathcal{D}} k(\cdot, x) \tag{11}$$

Thanks to "the kernel trick," we can make the kernel embeddings $\phi(x) = k(\cdot, x)$ of (11) disappear via the reproducing property, $\langle k(\cdot, x), k(\cdot, y) \rangle_{\mathcal{H}} = k(x, y)$.

Expanding out the squared norm of (11) as an inner product, we get

$$
\begin{aligned}
\mathrm{var}(\mathcal{D}) &= \frac{1}{|\mathcal{D}|} \sum_{x \in \mathcal{D}} \left\| k(\cdot, x) - \frac{1}{|\mathcal{D}|} \sum_{y \in \mathcal{D}} k(\cdot, y) \right\|_{\mathcal{H}}^2 = \frac{1}{|\mathcal{D}|} \sum_{x \in \mathcal{D}} \| k(\cdot, x) - \widehat{\mu}_k(\mathcal{D}) \|_{\mathcal{H}}^2 \\
&= \frac{1}{|\mathcal{D}|} \sum_{x \in \mathcal{D}} \langle k(\cdot, x) - \widehat{\mu}_k(\mathcal{D}), k(\cdot, x) - \widehat{\mu}_k(\mathcal{D}) \rangle_{\mathcal{H}} \\
&= \frac{1}{|\mathcal{D}|} \sum_{x \in \mathcal{D}} \left( \langle k(\cdot, x), k(\cdot, x) \rangle_{\mathcal{H}} - 2 \langle k(\cdot, x), \widehat{\mu}_k(\mathcal{D}) \rangle_{\mathcal{H}} + \| \widehat{\mu}_k(\mathcal{D}) \|_{\mathcal{H}}^2 \right) \\
&= \left( \frac{1}{|\mathcal{D}|} \sum_{x \in \mathcal{D}} k(x, x) \right) - 2 \left\langle \frac{1}{|\mathcal{D}|} \sum_{x \in \mathcal{D}} k(\cdot, x), \widehat{\mu}_k(\mathcal{D}) \right\rangle_{\mathcal{H}} + \left( \frac{1}{|\mathcal{D}|} \sum_{x \in \mathcal{D}} \| \widehat{\mu}_k(\mathcal{D}) \|_{\mathcal{H}}^2 \right) \\
&= \left( \frac{1}{|\mathcal{D}|} \sum_{x \in \mathcal{D}} k(x, x) \right) - 2 \| \widehat{\mu}_k(\mathcal{D}) \|_{\mathcal{H}}^2 + \| \widehat{\mu}_k(\mathcal{D}) \|_{\mathcal{H}}^2 \\
&= \frac{1}{|\mathcal{D}|} \sum_{x \in \mathcal{D}} k(x, x) - \left\langle \frac{1}{|\mathcal{D}|} \sum_{x \in \mathcal{D}} k(\cdot, x), \frac{1}{|\mathcal{D}|} \sum_{y \in \mathcal{D}} k(\cdot, y) \right\rangle_{\mathcal{H}} \\
&= \frac{1}{|\mathcal{D}|} \sum_{x \in \mathcal{D}} k(x, x) - \frac{1}{|\mathcal{D}|^2} \sum_{x \in \mathcal{D}} \sum_{y \in \mathcal{D}} k(x, y)
\end{aligned}
$$

via the reproducing property, linearity of the inner product, and symmetry of the positive-definite kernel $k$.

Other ways of looking at $\mathrm{var}(\mathcal{D})$ are the trace of the covariance operator and as part of the MMD. The latter is most relevant here. The cost $\mathrm{var}(\mathcal{D})$ only measures the "within $\mathcal{D}$" variance. If you partition $\mathcal{D}$ into two disjoint sets $\mathcal{D}_1$ and $\mathcal{D}_2$ such that $\mathcal{D} = \mathcal{D}_1 \cup \mathcal{D}_2$, then the reduction in this variance cost $\mathrm{var}(\mathcal{D}) - (\mathrm{var}(\mathcal{D}_1) + \mathrm{var}(\mathcal{D}_2))$ is directly proportional to the MMD.

The empirical MMD between two bags of samples $\mathcal{D}_1$ and $\mathcal{D}_2$ is defined as follows:

$$
\widehat{MMD}^2(\mathcal{D}_1, \mathcal{D}_2) = \| \widehat{\mu}_k(\mathcal{D}_1) - \widehat{\mu}_k(\mathcal{D}_2) \|_{\mathcal{H}}^2.
$$

Doing the same kind of inner product reproducing property trickery as above, we get the following equivalent expression:

$$
\widehat{MMD}^2(\mathcal{D}_1, \mathcal{D}_2) = \frac{1}{|\mathcal{D}_1|^2} \sum_{x \in \mathcal{D}_1} \sum_{y \in \mathcal{D}_1} k(x, y) + \frac{1}{|\mathcal{D}_2|^2} \sum_{x \in \mathcal{D}_2} \sum_{y \in \mathcal{D}_2} k(x, y) - \frac{2}{|\mathcal{D}_1||\mathcal{D}_2|} \sum_{x \in \mathcal{D}_1} \sum_{y \in \mathcal{D}_2} k(x, y)
$$

Define the cost function as follows:

$$
\boxed{\mathrm{cost}(\mathcal{D}) = |\mathcal{D}| \cdot \mathrm{var}(\mathcal{D})}. \tag{12}
$$

Then the difference between the cost of one bag of samples (i.e., with no change point) $\mathcal{D}$ and the sum of the costs of two disjoint bags $\mathcal{D}_1$ and $\mathcal{D}_2$ such that $\mathcal{D} = \mathcal{D}_1 \cup \mathcal{D}_2$ (i.e., one change point) is proportional to the empirical estimate of the MMD, $\widehat{MMD}^2(\mathcal{D}_1, \mathcal{D}_2)$:

$$
\begin{aligned}
\mathrm{cost}(\mathcal{D}) - (\mathrm{cost}(\mathcal{D}_1) + \mathrm{cost}(\mathcal{D}_2)) &= \left( \sum_{x \in \mathcal{D}} k(x, x) - \left( \sum_{x \in \mathcal{D}_1} k(x, x) + \sum_{x \in \mathcal{D}_2} k(x, x) \right) \right) + \\
&\quad \left( \left( \frac{1}{|\mathcal{D}_1|} \sum_{x \in \mathcal{D}_1} \sum_{y \in \mathcal{D}_1} k(x, y) + \frac{1}{|\mathcal{D}_2|} \sum_{x \in \mathcal{D}_2} \sum_{y \in \mathcal{D}_2} k(x, y) \right) - \frac{1}{|\mathcal{D}|} \sum_{x \in \mathcal{D}} \sum_{y \in \mathcal{D}} k(x, y) \right) \\
&= |\mathcal{D}_1| \cdot \| \widehat{\mu}_k(\mathcal{D}_1) \|_{\mathcal{H}}^2 + |\mathcal{D}_2| \cdot \| \widehat{\mu}_k(\mathcal{D}_2) \|_{\mathcal{H}}^2 - |\mathcal{D}| \cdot \| \widehat{\mu}_k(\mathcal{D}) \|_{\mathcal{H}}^2 \\
&= \frac{|\mathcal{D}_1| \cdot |\mathcal{D}_2|}{|\mathcal{D}|} \| \widehat{\mu}_k(\mathcal{D}_1) - \widehat{\mu}_k(\mathcal{D}_2) \|_{\mathcal{H}}^2 \\
&= \frac{|\mathcal{D}_1| \cdot |\mathcal{D}_2|}{|\mathcal{D}|} \widehat{MMD}^2(\mathcal{D}_1, \mathcal{D}_2),
\end{aligned}
$$

since

$$\widehat{\mu}_k(\mathcal{D}) = \frac{1}{|\mathcal{D}|} \sum_{x \in \mathcal{D}} k(\cdot, x) = \frac{1}{|\mathcal{D}|} \left( \sum_{x \in \mathcal{D}_1} k(\cdot, x) + \sum_{x \in \mathcal{D}_2} k(\cdot, x) \right) = \frac{|\mathcal{D}_1| \cdot \widehat{\mu}_k(\mathcal{D}_1) + |\mathcal{D}_2| \cdot \widehat{\mu}_k(\mathcal{D}_2)}{|\mathcal{D}|}.$$

None of these calculations change when each $x \in \mathcal{D}_1$ and $y \in \mathcal{D}_2$ are chunks rather than individual samples. We use the same cost function (12). The difference between the cost of one big bag $\mathcal{D}$ and two smaller bags $\mathcal{D}_1$ and $\mathcal{D}_2$ that result when $\mathcal{D}$ is partitioned by a change point is still proportional to $\widehat{MMD}^2(\mathcal{D}_1, \mathcal{D}_2)$. If the chunking stride is the chunk size (i.e., we partition the signal into chunks), we introduce no dependence issues; there are just fewer chunks in each bag (in exchange for the ability to make spectrally informative comparisons between chunks). None of the consistency results change.

It is not surprising, but worth checking, that, even when we use overlapping chunks, an algorithm like PELT will remain consistent.

**Proposition 1.** *We consider the regime where the chunk size $T$ is fixed, but the number of chunks $N$ in a segment (region between splits) grows as the segment length $NT \to \infty$. Let $\mathbf{X}_i = (x_i, \ldots, x_{i+T-1}) \in \mathcal{X}^T$ be the $i$-th chunk and $k$ a characteristic graph signal kernel $k : \mathcal{X}^T \times \mathcal{X}^T \to \mathbb{R}$ with RKHS $\mathcal{H}$. Let $P$ be the population distribution of a chunk. If the original time series is a stationary process within a segment, $P$ is the joint distribution of $T$ consecutive observations. For a segment $\mathcal{D} = \{\mathbf{X}_1, \ldots, \mathbf{X}_N\}$ of $N$ chunks, the average empirical cost*

$$\frac{1}{N}\text{cost}(\mathcal{D}) = \frac{1}{N} \sum_{i=1}^{N} k(\mathbf{X}_i, \mathbf{X}_i) - \frac{1}{N^2} \sum_{i=1}^{N} \sum_{j=1}^{N} k(\mathbf{X}_i, \mathbf{X}_j) = \frac{1}{N} \sum_i \|\phi(\mathbf{X}_i) - \widehat{\mu}_k\|_{\mathcal{H}}^2$$

*converges to the population variance.*

*Proof.* We suppose that the original signal is i.i.d., so that the chunks $\{\mathbf{X}_i\}$ are constructed from a sliding window of size $T$, the sequence is $m$-dependent with $m = T - 1$ (i.e., $\mathbf{X}_i$ and $\mathbf{X}_j$ are independent if $|i - j| \geq T$). (The argument continues to hold with standard mixing assumptions in the literature, such as $\phi$-mixing with geometric decay.)

Let $\phi(\mathbf{X}_i) = k(\cdot, \mathbf{X}_i)$ be the feature map and $\{\zeta_i\}$ the sequence of centered embeddings, i.e., $\zeta_i = \phi(\mathbf{X}_i) - \mu_k(P)$. Thus,

$$\frac{1}{N} \sum_{i=1}^{N} \zeta_i = \widehat{\mu_k}(\mathcal{D}) - \mu_k(P).$$

The variance of the empirical mean embedding is given by the following:

$$\mathbb{E}_P \|\widehat{\mu}_k(\mathcal{D}) - \mu_k(P)\|_{\mathcal{H}}^2 = \mathbb{E}_P \left\| \frac{1}{N} \sum_{i=1}^{N} \zeta_i \right\|_{\mathcal{H}}^2 = \frac{1}{N^2} \sum_{i=1}^{N} \sum_{j=1}^{N} \mathbb{E}_P \langle \zeta_i, \zeta_j \rangle_{\mathcal{H}}. \tag{13}$$

By $m$-dependence, $\mathbb{E}_P \langle \zeta_i, \zeta_j \rangle_{\mathcal{H}} = 0$ for $|i - j| \geq T$. The number of non-zero terms in the double sum of (13) is at most $N(2T - 1)$. Since the kernel is bounded, there is a $\sigma^2$ such that $\mathbb{E}_P \|\zeta_i\|^2 \leq \sigma^2$. Thus, we have:

$$\mathbb{E}_P \|\widehat{\mu}_k(\mathcal{D}) - \mu_k(P)\|_{\mathcal{H}}^2 \leq \frac{N(2T - 1)\sigma^2}{N^2} = O\left(\frac{T}{N}\right). \tag{14}$$

By Chebyshev's inequality, this implies $\|\widehat{\mu}_k(\mathcal{D}) - \mu_k(P)\|_{\mathcal{H}} = O_p(\sqrt{T/N})$.

Finally, by the Law of Large Numbers for $m$-dependent sequences, the average kernel diagonal $\frac{1}{N} \sum k(\mathbf{X}_i, \mathbf{X}_i)$ converges to $\mathbb{E}_P[k(\mathbf{X}, \mathbf{X})]$ almost surely. With $\widehat{\mu}_k \to_p \mu_k$, the continuous mapping theorem yields $\frac{1}{N}\text{cost} \to_p \text{var}_P(\phi(\mathbf{X}))$.

$\square$

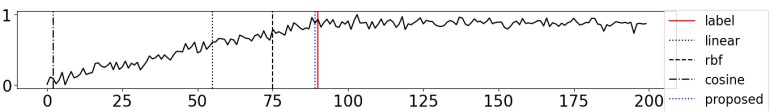

*Figure 8.* A signal (length 200) has a shift in slope at location $t = 90$. The proposed model –having Laplace option 'tps', linear kernel with chunk size 5 – can fairly detect the true changepoint while the baseline ones fail.

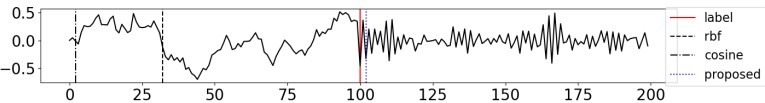

*Figure 9.* A signal (length 200) has a change in correlation at location $t = 100$. The proposed model –having Laplace option 'h1', linear kernel with chunk size 15 – can fairly detect the true changepoint while the baseline ones fail.

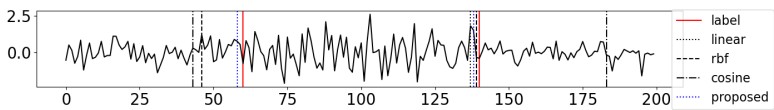

*Figure 10.* A signal (length 200) mean 0 having 2 changes in variance at location $t = 60$ and $140$. The proposed model –having Laplace option 'h1', linear kernel with chunk size 15 – can fairly detect the true changepoints as good as the state of the art ones.

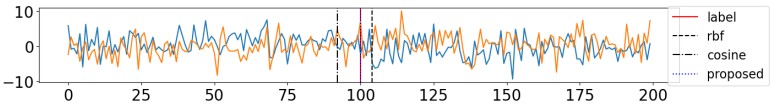

*Figure 11.* A 2D signal (length 200) has one changepoint in mean at location $t = 100$. The proposed model –having Laplace option 'heat', linear kernel with chunk size 2 – can fairly detect the true changepoint as good as the state of the art ones.

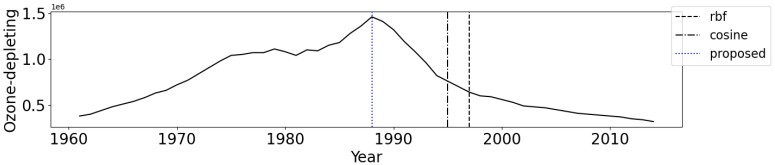

*Figure 12.* Levels of ozonedepleting substance in the atmosphere. The Montreal Protocol came into force in September 1989. Data from www.ourworldindata.

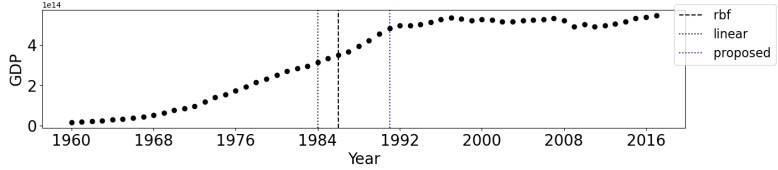

*Figure 13.* The GDP of Japan in constant local currency. Data obtained from the World Bank.

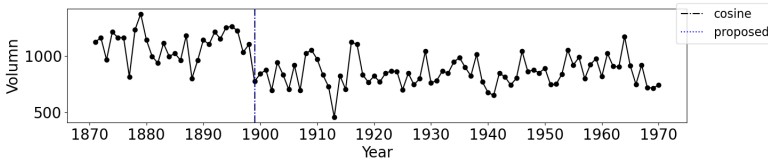

*Figure 14.* Yearly volume of the Nile river at Aswan. A dam was built in 1898. Data obtained from the website for the book by Durbin and Koopman (2012).

## F. Additional Experiments

## G. Dirichlet Energy Calculation

The calculation follows the well-known scalar case:

$$E(f) = \sum_{\substack{e \in E \\ e=\{i,j\}}} w_{i,j} \|f(i) - f(j)\|_{\mathcal{D}}^2$$

$$= \sum_{\substack{e \in E \\ e=\{i,j\}}} w_{i,j}(\|f(i)\|_{\mathcal{D}}^2 + \|f(j)\|_{\mathcal{D}}^2) - \sum_{\substack{e \in E \\ e=\{i,j\}}} 2w_{i,j}\langle f(i), f(j)\rangle_{\mathcal{D}}$$

$$= \sum_{i=1}^{T} \|f(i)\|_{\mathcal{D}}^2 \left(\sum_{j=1}^{T} w_{i,j}\right) - \sum_{i=1}^{T}\sum_{j=1}^{T} 1 \cdot w_{i,j}\langle f(i), f(j)\rangle_{\mathcal{D}}$$

$$= \sum_{i=1}^{T} \mathbf{D}_{i,i}\|f(i)\|_{\mathcal{D}}^2 - \sum_{i=1}^{T}\sum_{j=1}^{T} \mathbf{A}_{i,j}\langle f(i), f(j)\rangle_{\mathcal{D}}$$

$$= \sum_{i=1}^{T}\sum_{j=1}^{T} \mathbf{D}_{i,j}\langle f(i), f(j)\rangle_{\mathcal{D}} - \sum_{i=1}^{T}\sum_{j=1}^{T} \mathbf{A}_{i,j}\langle f(i), f(j)\rangle_{\mathcal{D}}$$

$$= \sum_{i=1}^{T}\sum_{j=1}^{T} \mathbf{L}_{i,j}\langle f(i), f(j)\rangle_{\mathcal{D}} = \langle \mathbf{L}, \mathbf{G}\rangle_F,$$

where $\mathbf{G}$ is the Gram matrix of $\langle\cdot,\cdot\rangle_{\mathcal{D}}$ on the functions $\{f(i)\}_{i=1}^{T}$, satisfying $\mathbf{G}_{i,j} = \langle f(i), f(j)\rangle_{\mathcal{D}}$.

## H. Code

For test signals generation:

```
n_signals = 500
signal_length = 200
n_per_type = int(n_signals/5)

signals = []
true_bkps = []
signal_types = []

for signal_type in ["mean", "variance", "frequency", "slope", "correlation"]:
    for _ in range(n_per_type):
        shift = np.random.uniform(0.03, 1.0)
        y = np.zeros(signal_length)
        t_star = np.random.randint(80, 121)

        if signal_type == "mean":
            sd = np.random.uniform(2, 4)
            y[:t_star] = np.random.normal(-1, sd, t_star)
            y[t_star:] = np.random.normal(1, sd, signal_length - t_star)

        elif signal_type == "variance":
            v = np.random.uniform(0.8, 1.2)
            y[:t_star] = np.random.normal(0, np.sqrt(v), t_star)
            y[t_star:] = np.random.normal(0, np.sqrt(v + shift), signal_length - t_star)

        elif signal_type == "frequency":
```

```
1320              t = np.arange(t_star)
1321              y[:t_star] = np.cos(2 * np.pi * 0.05 * t)
1322              t2 = np.arange(signal_length - t_star)
1323              y[t_star:] = np.cos(2 * np.pi * (0.05 + shift) * t2)
1324              y = y + np.random.normal(0, 0.1, signal_length)
1325
1326          elif signal_type == "slope":
1327              slope_before = 0.2
1328              y[:t_star] = np.linspace(0, 1, t_star) * slope_before
1329              y[t_star:] = np.linspace(0, 1, signal_length - t_star) * (slope_before + shift)
1330              y = y + np.random.normal(0, 0.1, signal_length)
1331
1332          elif signal_type == "correlation":
1333              f = np.random.uniform(0.3, 0.9)
1334              y[0]=0
1335              for t in range(1, signal_length):
1336                  if t < t_star:
1337                      y[t] = f * y[t-1]
1338                  else:
1339                      y[t] = -f * y[t-1]
1340              y = y + np.random.normal(0, 0.1, signal_length)
1341
1342          signals.append(y)
1343          true_bkps.append(t_star)
1344          signal_types.append(signal_type)
1345
1346
1347 def detect_all_models(y):
1348     out = {}
1349     out["linear"] = rpt.KernelCPD(kernel="linear").fit(y).predict(n_bkps=1)[0]
1350     out["cosine"] = rpt.KernelCPD(kernel="cosine").fit(y).predict(n_bkps=1)[0]
1351     out["rbf"] = rpt.KernelCPD(kernel="rbf").fit(y).predict(n_bkps=1)[0]
1352     out["proposed"] = (
1353             binseg(
1354                 sequence=y,
1355                 n_changepoints=1,
1356                 T=np.random.randint(2, 10),
1357                 laplace_option=np.random.choice(["heat", "dirichlet", "tps"]),
1358                 laplace_hyperparameter=np.random.randint(1, 2)
1359             )[0]
1360         )
1361     return out
1362
1363 detections = Parallel(n_jobs=-1)(delayed(detect_all_models)(y) for y in signals)
1364
1365 acc_df = pd.DataFrame(
1366     0.00,
1367     index=["linear", "cosine", "rbf", "proposed"],
1368     columns=["mean", "variance", "frequency", "slope", "correlation"]
1369 )
1370
1371 for model in ["linear", "cosine", "rbf", "proposed"]:
1372     for stype in ["mean", "variance", "frequency", "slope", "correlation"]:
1373         correct = 0
1374
```

```
1375        for i in range(len(signals)):
1376            if signal_types[i] != stype:
1377                continue
1378            cp = detections[i][model]
1379            if cp >= 0 and abs(cp - true_bkps[i]) <= 5:
1380                correct += 1
1381    acc_df.loc[model, stype] = correct / n_per_type
1382
1383
1384
1385
1386
1387
1388
1389
1390
1391
1392
1393
1394
1395
1396
1397
1398
1399
1400
1401
1402
1403
1404
1405
1406
1407
1408
1409
1410
1411
1412
1413
1414
1415
1416
1417
1418
1419
1420
1421
1422
1423
1424
1425
1426
1427
1428
1429
```

```
        for i in range(len(signals)):
            if signal_types[i] != stype:
                continue
            cp = detections[i][model]
            if cp >= 0 and abs(cp - true_bkps[i]) <= 5:
                correct += 1
    acc_df.loc[model, stype] = correct / n_per_type
```