# OpenReview forum: "Temporally Aware Kernel Change-Point Detection with Graph Function Kernels"
_ICML.cc/2026/Conference — Submitted to ICML 2026_

### Official Review · Reviewer_YPzP · 2026-03-09

**Soundness:** 2
**Presentation:** 2
**Significance:** 1
**Originality:** 3
**Overall Recommendation:** 1
**Confidence:** 5

**Summary:**

This paper proposes a temporally aware kernel framework for change-point detection in time series that incorporates the temporal or graph structure of observations rather than treating samples as unordered sets. It introduces graph signal kernels that compare chunks of time series using covariance or graph Laplacian structure, enabling detection of spectral changes such as shifts in frequency or autocorrelation.

**Compliance With Llm Reviewing Policy:**

Affirmed.

**Final Justification:**

There was not response to my comments.

**Key Questions For Authors:**

1- Can the authors provide a more rigorous and comprehensive simulation study? In particular, it would be helpful to clearly specify the data-generating mechanisms, consider different scaling regimes, compare against a broad set of competing methods, and report multiple evaluation metrics such as Hausdorff distance and change-point estimation error. The study should also examine scenarios with multiple change points, varying levels of separation between change points, and repeated Monte Carlo runs, while clearly explaining how tuning parameters are selected for each method.  The author have the attitude that it is the job of the reader to validate the proposed method!

2- The consistency result presented in the appendix appears weaker than the theoretical guarantees typically established in the statistical and machine learning literature on change-point detection. In particular, it would be valuable for the authors to provide a more complete theoretical analysis, such as establishing localization rates for estimating the positions of multiple change points under the proposed model. See:

 "Optimal nonparametric multivariate change point detection and localization
OHM Padilla, Y Yu, D Wang, A Rinaldo
IEEE Transactions on Information Theory 68 (3), 1922-1944"

3- Why do the authors omit several relevant references from the literature on nonparametric change-point detection? The paper would benefit from a more comprehensive discussion of related work, including studies such as the following:

"Song, Hoseung, and Hao Chen. "Practical and powerful kernel-based change-point detection." IEEE Transactions on Signal Processing 72 (2024): 5174-5186."

"C.M. Madrid-Padilla, H. Xu, Daren Wang, O.H. Madrid-Padilla, Y. Yu. Change point detection and inference in multivariable nonparametric models under mixing conditions. Advances in Neural Information Processing Systems (NeurIPS), 36: 21081-21134, 2023"

 "Optimal nonparametric multivariate change point detection and localization
OHM Padilla, Y Yu, D Wang, A Rinaldo
IEEE Transactions on Information Theory 68 (3), 1922-1944"

"O.-H. Madrid-Padilla, Yi Yu, Daren Wang, Alessandro Rinaldo. Optimal nonparametric change point detection and localization. Electronic Journal of Statistics. 15 (1) 1154 - 1201, 2021"

4- How does the theoretical analysis of the proposed method compare with that of  "Song, Hoseung, and Hao Chen. "Practical and powerful kernel-based change-point detection. IEEE Transactions on Signal Processing 72 (2024): 5174-5186"? In particular, are there differences in assumptions, guarantees, or detection/localization rates? Additionally, how does the proposed method compare empirically with this approach in terms of detection accuracy and computational performance?

5-What is the computational cost of the proposed method?

**Limitations:**

No. Please discuss under what mathematical conditions the proposed method is expected to perform well.

**Strengths And Weaknesses:**

Strengths

1- The idea seems novel.


Weaknesses
1- There is no rigorous empirical evaluation.
2- The theoretical results are lacking.
3-Poorly written. For example, there are equations with "?".

---

### Official Review · Reviewer_MmLb · 2026-03-11

**Soundness:** 2
**Presentation:** 2
**Significance:** 2
**Originality:** 3
**Overall Recommendation:** 2
**Confidence:** 4

**Summary:**

The paper proposes a family of graph-based kernels for changepoint detection in time series. Their method consists of a Dirichlet kernel that is built from the time information in the time series and the values of the time series themselves. Experimental results are provided to support some of their claims.

**Compliance With Llm Reviewing Policy:**

Affirmed.

**Final Justification:**

Since there was no response from the authors, I will maintain my score.

**Key Questions For Authors:**

1. How does your work compare with global alignment kernel (GAK) or a time series kernel like signature kernel?

2. What are the computational requirements of your method? For large time series, for example, the computation could be prohibitively high.

3. How are the kernel’s hyperparameters tuned?

**Limitations:**

Yes

**Strengths And Weaknesses:**

Soundness: The work is technically sound. They discuss the proposed method clearly – the kernels themselves, and the induced distance metrics. What is lacking is a strong experimental section.  This section is short on details such as: What is the data used, what changepoint method was used?, what are the baselines, and why? etc.

Presentation: The proposed methods are clearly described, and the paper in general is easy to read. There are however some missing information in Section 5 (see previous comments on soundness). There are some missing references (? in 2 places).

Significance: The proposed method could potentially be an improvement on changepoint detection. But the lack of solid experimental verification of the claims makes it a bit weak.

Originality: The proposed method is moderately original, in that, they provide some insight into how graph based kernels can be used in time series applications. However, a comparison between their method and kernels like the GAK (global alignment kernel) or the signature kernel for time series has not been done.

---

### Decision · Program_Chairs · 2026-04-30

**Decision:**

Reject

**Comment:**

The paper presents a temporally aware kernel framework for change-point detection in time series. The review finds the paper is not ready for the conference, raising comprehensive concerns, including the soundness of the design and settings, the theoretical analysis and justification of the design and performance, the valid experiments and evaluation, in the SOTA context.